# A Mechanism of Implicit Regularization in Deep Learning

## Abstract

Despite a lot of theoretical efforts, very little is known about mechanisms of implicit regularization by which the low complexity contributes to generalization in deep learning. In particular, causality between the *generalization performance*, *implicit regularization* and *nonlinearity of activation functions* is one of the basic mysteries of deep neural networks (DNNs). In this work, we introduce a novel technique for DNNs called **"random walk analysis"** and reveal a mechanism of the implicit regularization caused by nonlinearity of ReLU activation. Surprisingly, our theoretical results suggest that the learned DNNs interpolate almost linearly between data points, which leads to the low complexity solutions in the over-parameterized regime. As a result, we prove that stochastic gradient descent can learn a class of continuously differentiable functions with generalization bounds of the order of $O(n^{-2})$ ($n$: the number of samples). Furthermore, our analysis is independent of the kernel methods, including neural tangent kernels.

## 1 Introduction

Deep Neural Networks (DNNs) have demonstrated dominating performance in numerous machine learning tasks, and it shows great generalization performance in the over-parameterized regime. Theoretically, mechanisms of implicit regularization, which is considered as an important factor of such generalization performance in over-parameterized DNNs, still remain unknown (Zhang et al., 2016; Neyshabur, 2017). In the over-parameterized regime, recent studies report that generalization bounds for DNNs can be obtained by replacing the neural network with its linear approximation model with respect to weight parameters at initialization. Most of these studies rely on the connection between deep learning and neural tangent kernels (NTKs) (Daniely et al., 2016; Jacot et al., 2018; Arora et al., 2019b), which characterizes the dynamics of network outputs throughout gradient descent training in the infinite width limit. However, a source of implicit regularization in over-parameterized DNNs has not been identified. Recent empirical and theoretical results indicate that generalization performance and implicit regularization of over-parameterized DNNs cannot be captured by NTK analysis (Wei et al., 2018; Allen-Zhu & Li, 2019; Woodworth et al., 2019; Geiger et al., 2019). Understanding how implicit regularization properly controls the superfluous expressive power of over-parameterized DNNs gives us new insights into the theoretical analysis in deep learning. This leads to the first question:

**Question 1.** *What kind of low complexity is caused by implicit regularization in deep learning?*

Linear networks without activation functions are important subject, and there are a number of theoretical works on the implicit regularization in over-parameterized neural networks mainly focusing on *linear* models (Ji & Telgarsky, 2018; Gidel et al., 2019; Arora et al., 2019a). In contrast, whole properties of over-parameterized DNNs that may result from nonlinearity of activation functions cannot be captured by the approximated linear models, specifically, the kernel regression predictor using the NTK. However, some mechanisms of the implicit regularization can depend on *nonlinearity*. This leads to the next question:

**Question 2.** *Can we identify a mechanism of implicit regularization that depends on nonlinearity of activation functions?*

Now for optimization algorithms, the training dynamics of full batch gradient descent (GD) is better understood although GD is too expensive for most applications and one often uses stochastic gra-

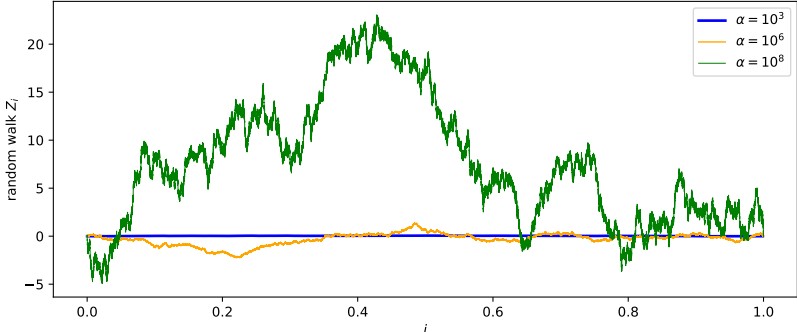

Figure 1: Random walks having a step size distributed according to $\mathcal{N}(0, 2/m)$ ($m = 10^3$) after $\alpha$ steps. The smaller the number of steps $\alpha$ is, the more straight. For visibility, we transformed the random walk sequence $\{y_i\}_{i=0}^{\alpha}$ into $\{z_i\}_{i=0}^{\alpha}$ defined by $z_i := (y_i - y_0) - (i/\alpha)(y_\alpha - y_0)$.

dient decent (SGD) instead. In most cases, the authors used GD to derive their results by the NTK analysis. Recent work showed that without any structural assumptions about the data distribution, two-layer or three-layer over-parameterized networks trained by SGD can learn $C^\infty$-class functions (Allen-Zhu et al., 2018a; Arora et al., 2019c). However, it is not clear the relation between generalization and implicit regularization for DNNs optimized by SGD. Towards this end, the following question is also unsolved:

**Question 3.** *Is it possible to obtain provable generalization bounds based on the implicit regularization for DNNs optimized by SGD?*

To answer these questions, we introduce a novel analysis for DNNs and characterize a mechanism of implicit regularization that caused by nonlinearity of ReLU activation. Our results indicate that the DNNs (trained by SGD) interpolate almost linearly between data points, which leads to the low complexity solutions in the over-parameterized regime. Accordingly, we prove that SGD with random initialization can learn a class of continuously differentiable functions with generalization error bounds of the order of $O(n^{-2})$ ($n$: the number of samples), which is independent of the NTK analysis.

In order to introduce our analysis for implicit regularization, let us focus on the DNN output on a one-dimensional linear path between training points. We define it as $\boldsymbol{x}(s) := (1 - s)\boldsymbol{x}^{(p)} + s\boldsymbol{x}^{(q)}$ ($s \in [0, 1]$), where $\boldsymbol{x}^{(p)}$ and $\boldsymbol{x}^{(q)}$ are training points. The corresponding function of the DNN with ReLU (ReLU DNN) is continuous piecewise linear on the linear path $\boldsymbol{x}(s)$. In the hidden layer, the corresponding function $g^l(\boldsymbol{x})$ of each unit in the $l$-th layer is also continuous piecewise linear on the linear path $\boldsymbol{x}(s)$, that is, a composite function $(g^l \circ \boldsymbol{x})(s)$ is continuous piecewise linear. Since DNN nonlinearity is linked to breakpoints, which is caused by ReLU activation, the set of the breakpoints plays a key role in the network behavior. These kind of breakpoints are also known as *kinks* or *knots* (Steinwart, 2019). Since $(g^l \circ \boldsymbol{x})(s)$ is continuous piecewise linear, the gradient of $(g^l \circ \boldsymbol{x})(s)$ has gaps at the breakpoints, which we call *gradient gaps*. We focus on gradient gaps with respect to the parameter $s$.[1] Our key finding is that the gradient gaps are a constant multiple of independent Gaussian random variables according to $\mathcal{N}(0, 2/m)$, where $m$ is the number of units in the hidden layer. Since the gradient of the unit $(g^l \circ \boldsymbol{x})(s)$ is the sum of the gradient gaps, it is a "Gaussian random walk"[2] (see Step 2 in the proof of Lemma 3). The relation between the step size variance and the number of steps determines the Gaussian random walk behavior. In the corresponding function of the DNN, the step size variance is the variance of weights, and the number of steps is the number of breakpoints. In a regime where ( the number of steps $\times$ the step size variance) $\leq O(1)$, the Gaussian random walk strolls little from the origin with high probability (see Figure 1). Our results show that the ReLU DNN is in the same regime and "simplicity" of the gradient depends crucially on the number of breakpoints. Hanin & Rolnick (2019) proves that the average number of breakpoints

---

[1] It is worth pointing out that in this paper, each gradient is a derivative not with respect to *weight parameters* but with respect to *input parameters*.

[2] If $\{X_i\}_{i=1}^{\alpha}$ are independent Gaussian random variables, then $\{Y_i\}_{i=1}^{\alpha}$, where $Y_i := X_1 + X_2 + \cdots + X_i$, is called a Gaussian random walk.

is linear in the number of hidden units at initialization, and we also give a proof of an upper bound on the number of breakpoints even after training (see §3.1).

In this work, we prove *a priori* generalization estimates, which is independent of the posterior data distribution, by analyzing the behavior of DNNs on the linear path $\boldsymbol{x}(s)$. In other words, we show that in Theorem 2, the difference between the network output function and its linear interpolation is evaluated only from the amount of the weight change. We note that in Theorem 2, we dose not use the properties of the trained neural networks. Our key analysis is that the gradient of unit $(g^l \circ \boldsymbol{x})(s)$ is a "Gaussian random walk" and approximately equal to a straight line on the linear path $\boldsymbol{x}(s)$, which we call **random walk analysis**. Our technique is based on *a priori* estimates that variation of the gradients between data points is extremely small and depends essentially on the value of the number of breakpoints times the variance of weights.

**Difference from NTK.** Interestingly, our analysis idea is different from other previous work based on NTK. Our findings are some novel aspects as follows:

- The NTK is defined using the gradient of the DNN output with respect to **weight parameter space**. In contrast, the linear approximation (Lemma 3 in this paper) is defined using the gradient of the DNN output with respect to **input parameter space**. In other words, the variables to be differentiated are different.

- The random walk analysis indicates that over-parameterized ReLU DNNs interpolate almost linearly between the data points. For ReLU activation, since the NTK kernel mapping is not Lipschitz but $1/2$-Hölder, it is difficult to obtain such a result in the NTK analysis without a tradeoff between smoothness and approximation (Bietti & Mairal, 2019).

**Our Contributions.** In this work, we consider an $L$-layer over-parameterized ReLU neural network with $l^2$ regression task using SGD from random initialization. We show the constitutive relation between implicit regularization and generalization, which enables us to provide new insights on the role of implicit regularization in deep learning. Our main contributions are as follows:

- Our random walk analysis provides *a priori* estimates of low complexity in over-parameterized deep neural networks, and directly indicates that the unit output between data points is properly controlled by weight initialization and SGD to keep connecting the data points almost straight, which is one of the underlying mechanisms of implicit regularization in the over-parameterized regime.

- Our result suggests that implicit regularization is attributed to the nonlinearity of ReLU DNN, which is indicated by the fact that variance of weight and the number of breakpoints determine the Gaussian random walk behavior.

- We also prove that in one-dimensional input case, SGD with random initialization can learn $C^1$-class functions. Our generalization estimates are based on the implicit regularization.

## 2 PRELIMINARIES AND NOTATION

**Notation.** For $n \in \mathbb{N}$, we let $[n] = \{1, 2, \ldots, n\}$. We use $\mathcal{N}(\mu, \sigma^2)$ to denote the Gaussian distribution of mean $\mu$ and variance $\sigma^2$. We use $\|\boldsymbol{v}\|_2$ to denote the Euclidean norm of a vector $\boldsymbol{v}$, use $\|\boldsymbol{v}\|_F$ to denote the Frobenius norm of a vector $\boldsymbol{v}$. When z is a sub-Gaussian random variable, we let $\|z\|_{\psi_2} = \inf\{t > 0 \mid \mathbb{E}[\exp(z^2/t^2)] \leq 2\}$ to denote sub-Gaussian norm (Vershynin, 2018). For a vector $\boldsymbol{v}$, we denote by $[\boldsymbol{v}]_i$ or $v_i$ the $i$-th element of $\boldsymbol{v}$. For a matrix $\boldsymbol{M}$, we denote by $[\boldsymbol{M}]_{i,j}$ or $M_{i,j}$ the entry in the $i$-th row and $j$-th column of $\boldsymbol{M}$, and we denote by $[\boldsymbol{M}]_i$ the $i$-th row vector of $\boldsymbol{M}$. We denote by $\mathbf{1}_{\{E\}}$ the indicator function for the event $E$. ReLU activation is given by $\phi(x) = \max\{0, x\}$, and for a vector $\boldsymbol{a} \in \mathbb{R}^n$, we define $\phi(\boldsymbol{a}) = \big(\phi(a_1), \phi(a_2), \ldots, \phi(a_n)\big)$.

**Network structure.** In this work, an $L$-layer fully-connected feed-forward ReLU neural network with $m$ units in each hidden layer is given by $f : \mathbb{R}^d \to \mathbb{R}^c$. For input $\boldsymbol{x} \in \mathbb{R}^d$, unit output $g^l(\boldsymbol{x})$ of each layer $l \in [L]$ and network output $f(\boldsymbol{x})$ are given by the following functions:

$$g^0(\boldsymbol{x}) := \mathbf{A}\boldsymbol{x}, \qquad g^l(\boldsymbol{x}) := \mathbf{W}^l\phi(g^{l-1}(\boldsymbol{x})) \ \ (l \in [L]), \qquad f(\boldsymbol{x}) := \mathbf{B}\phi(g^L(\boldsymbol{x})), \quad (1)$$

where $\mathbf{A} \in \mathbb{R}^{m \times d}$, $\mathbf{W}^l \in \mathbb{R}^{m \times m}$ $(l \in L)$, and $\mathbf{B} \in \mathbb{R}^{c \times m}$ are weight matrices. We assume the following Gaussian initialization: $A_{i,j} \sim \mathcal{N}(0, 2/m)$, $W^l_{i,j} \sim \mathcal{N}(0, 2/m)$, and $B_{i,j} \sim \mathcal{N}(0, 2/c)$. For input $\boldsymbol{x}$ and weight matrices $\overrightarrow{\mathbf{W}} := (\mathbf{W}^1, \mathbf{W}^2, \ldots, \mathbf{W}^L)$, the network output $f(\boldsymbol{x})$ is also denoted by $f(\overrightarrow{\mathbf{W}}, \boldsymbol{x})$. In this work, we only update weights in $\overrightarrow{\mathbf{W}}$ and leave $\mathbf{A}$ and $\mathbf{B}$ at the random initialization. For input $\boldsymbol{x} \in \mathbb{R}^d$ and $l \in [L]$, we denote by $\mathbf{G}^l(\boldsymbol{x})$ a diagonal matrix, which represents the activation pattern of the $l$-th layer, which we call an *indicator matrix*. More precisely, we define the $i$-th diagonal element as $\left[\mathbf{G}^l(\boldsymbol{x})\right]_{i,i} := \mathbf{1}_{\left\{g^l_i(\boldsymbol{x}) \geq 0\right\}}$. Since the ReLU activation is positive homogeneous, we obtain the following equality: $g^l(\boldsymbol{x}) = \mathbf{W}^l \mathbf{G}^{l-1}(\boldsymbol{x}) g^{l-1}(\boldsymbol{x})$.

**Dataset and loss function.** The data are generated from an unknown distribution $\mathcal{D}$ over $(\boldsymbol{x}, \boldsymbol{y}) \in \mathbb{R}^d \times \mathbb{R}^c$, where $\boldsymbol{x}$ is the input data point and $\boldsymbol{y}$ is the label associated with this data point. We assume without loss of generality that for each input $\boldsymbol{x} = (x_1, \ldots, x_d)$, using additional coordinates $(x_{d+1}, x_{d+2})$, the replacement input $\boldsymbol{x}'' := (x_1, \ldots, x_d, x_{d+1}, x_{d+2})$ is normalized so that $\|x''\|_2 = 1$ and its last coordinate $x_{d+2} = 1/\sqrt{2}$. [3] we also use $\boldsymbol{x}$ to denote the replacement input $\boldsymbol{x}'' \in \mathbb{R}^{d+2}$. The training data $\mathbb{Z} := \{(\boldsymbol{x}^{(1)}, \boldsymbol{y}^{(1)}), \ldots, (\boldsymbol{x}^{(n)}, \boldsymbol{y}^{(n)})\}$ is given as $n$ i.i.d. samples from $\mathcal{D}$. We define the minimum distance of the training data: $\delta := \min\{\|\boldsymbol{x}^{(i)} - \boldsymbol{x}^{(j)}\|_2 : \forall i, j \in [n], i \neq j\} > 0$.

For the $l^2$ regression loss $\ell(\hat{\boldsymbol{y}}, \boldsymbol{y}) := \frac{1}{2}\|\hat{\boldsymbol{y}} - \boldsymbol{y}\|^2_2$ and a subset of training data $\mathbb{Z}^{(\tau)} \subset \mathbb{Z}$, we define our regression objective as follows: $\mathcal{L}_{\mathbb{Z}^{(\tau)}}(\overrightarrow{\mathbf{W}}) := \mathbb{E}_{(\boldsymbol{x}, \boldsymbol{y}) \sim \mathbb{Z}^{(\tau)}}[\ell(f(\overrightarrow{\mathbf{W}}, \boldsymbol{x}), \boldsymbol{y})]$.

**Stochastic gradient descent with Gaussian initialization.** We use mini-batch SGD to train the network with a constant learning rate $\eta > 0$, a batch size $b$ and iteration number $T$. Let $\overrightarrow{\mathbf{W}}^{(0)} := (\mathbf{W}^{1,(0)}, \mathbf{W}^{2,(0)}, \ldots, \mathbf{W}^{L,(0)})$, $\mathbf{A}$, $\mathbf{B}$ be weight matrices generated from the above Gaussian initialization. Suppose we start at $\overrightarrow{\mathbf{W}}^{(0)}$ and for each $l \in [L]$ and $t = 0, 1, \ldots, T - 1$,

$$\mathbf{W}^{l,(t+1)} = \mathbf{W}^{l,(t)} - \eta \cdot \nabla_{\mathbf{W}^l} \mathcal{L}_{\mathbb{Z}^{(t)}}(\overrightarrow{\mathbf{W}}^{(t)}), \tag{2}$$

where $\mathbb{Z}^{(t)} \subset \mathbb{Z}$ is a mini-batch of size $b$. For input $\boldsymbol{x}$, each layer $l \in [L]$ and each step $t \in [T]$, we denote by $g^{l,(t)}(\boldsymbol{x})$ the unit output, $f^{(t)}(\boldsymbol{x})$ the network output, $\mathbf{W}^{l,(t)}$ the weight matrix, and $G^{l,(t)}(\boldsymbol{x})$ the indicator matrix.

In the above setting, recent paper (Allen-Zhu et al., 2018b) shows that SGD can allow an over-parameterized multi-layer network to attain arbitrarily low training error as follows:

**Theorem 1** (Convergence of SGD (Allen-Zhu et al., 2018b)). *For any* $\varepsilon \in (0, 1]$, $\delta \in (0, O(1/L)]$ *and* $b \in [n]$, *let* $m \geq \widetilde{\Omega}\left(\frac{\mathrm{poly}(n, L, \delta^{-1})d}{b}\right)$, $\eta := \Theta\left(\frac{b\delta d}{\mathrm{poly}(n,L)m \log^2 m}\right)$, $T = \Theta\left(\frac{\mathrm{poly}(n,L) \log^2 m}{b\delta^2} \log\left(\frac{n \log m}{\varepsilon}\right)\right)$, *and* $\overrightarrow{\mathbf{W}}^{(0)}, \mathbf{A}, \mathbf{B}$ *are at random initialization. Then, it satisfies with probability at least* $1 - e^{-\Omega(\log^2 m)}$ *over randomness of* $\mathbb{Z}^{(1)}, \ldots \mathbb{Z}^{(T)}$:
$\mathcal{L}_{\mathbb{Z}}(\overrightarrow{\mathbf{W}}) \leq \varepsilon$, $\quad \|\mathbf{W}^{l,(t)} - \mathbf{W}^{l,(0)}\|_F \leq O\left(\lambda \frac{\log m}{\sqrt{m}}\right)$, $\quad \|[\mathbf{W}^{l,(t)}]_i - [\mathbf{W}^{l,(0)}]_i\|_2 \leq O\left(\lambda \frac{\log m}{m}\right)$
$(\forall t \in [T])$, *where* $\quad \lambda := \frac{n^{3.5}\sqrt{c}}{\delta\sqrt{b}}$.

Our results on generalization also crucially depend on this analysis.

## 3 RANDOM WALK ANALYSIS

In this work, we consider the behavior of the unit output in each layer on a one-dimensional linear path between two data points, which we denote by $\boldsymbol{x}(s)$ ($s \in [0, 1]$). In this section, for notational simplicity, we drop the superscript with respect to $(t)$, which is the $t$-th iteration.

---

[3]Without loss of generality, one can rescale and assume $\|\boldsymbol{x}\|_2 \leq 1/\sqrt{2}$ for every input $\boldsymbol{x}$. Again, without loss of generality, one can pad each $\boldsymbol{x}$ by an additional coordinate $x_{d+1}$ to ensure $\|\boldsymbol{x}'\|_2 = 1/\sqrt{2}$. Finally, without loss of generality, one can pad each $\boldsymbol{x}'$ by an additional coordinate $x_{d+2}$ to ensure $\|\boldsymbol{x}''\|_2 = 1$. This last coordinate $x_{d+2} = 1/\sqrt{2}$ is equivalent to introducing a (random) bias term and this procedure is described in Allen-Zhu et al. (2018a).

**Definition 3.1** (One-dimensional Linear Path). For each pair of data points $\boldsymbol{x}^{(p)}$ and $\boldsymbol{x}^{(q)}$ $(p \neq q)$, We define $\boldsymbol{x}(s) := (1-s)\,\boldsymbol{x}^{(p)} + s\,\boldsymbol{x}^{(q)}$, $s \in [0,1]$, and denote by $\boldsymbol{v} := \boldsymbol{x}^{(q)} - \boldsymbol{x}^{(p)}$ the direction vector.

Note that the unit output $g_i^l(\boldsymbol{x}(s))$ and the network output $f_i(\boldsymbol{x}(s))$ are continuous piecewise linear functions on $s \in [0,1]$, and the network output can be expressed as follows:

$$f(\boldsymbol{x}(s)) = \mathbf{B}\mathbf{G}^L(\boldsymbol{x}(s))\mathbf{W}^L\mathbf{G}^{L-1}(\boldsymbol{x}(s))\mathbf{W}^{L-1}\cdots\mathbf{G}^1(\boldsymbol{x}(s))\mathbf{W}^1\mathbf{G}^0(\boldsymbol{x}(s))\mathbf{A}\boldsymbol{x}(s). \tag{3}$$

Note that $g_i^l(\boldsymbol{x}(s))$ and $f_i(\boldsymbol{x}(s))$ have linear approximations at the point $s = 0$ as follows[4]:

$$\tilde{g}_i^l(s) := g_i^l(\boldsymbol{x}(0)) + s \cdot \left[\frac{d}{ds}g_i^l(\boldsymbol{x}(s))\right]_{s=0}, \quad \tilde{f}_i(s) := f_i(\boldsymbol{x}(0)) + s \cdot \left[\frac{d}{ds}f_i(\boldsymbol{x}(s))\right]_{s=0}. \tag{4}$$

We prove that these piecewise linear functions are almost straight, which contributes to the generalization of NN. In other words, the unit output $g_i^l(\boldsymbol{x}(s))$ can be approximated by a linear function with high accuracy. Moreover, the small difference between $g_i^l(\boldsymbol{x}(s))$ and $\tilde{g}_i^l(s)$ indicates the low complexity of the unit output.

**Theorem 2** (A Priori Estimates for Implicit Regularization). *Under the same setting as Theorem 1, with probability at least $1 - e^{-\Omega(\log^2 m)}$, for every $\boldsymbol{x}^{(p)}$, $\boldsymbol{x}^{(q)}$ $(p,q \in [n]$, $p \neq q)$, $t \in [T]$, $l \in [L]$, $i \in [m]$ and $j \in [c]$, we have*

$$\sup_{0 \leq s \leq 1}\left|g_i^{l,(t)}(\boldsymbol{x}(s)) - \tilde{g}_i^{l,(t)}(s)\right| \leq O\left(\frac{\log^l m}{\sqrt{m}}\right)\|\boldsymbol{v}\|_2, \tag{5}$$

$$\sup_{0 \leq s \leq 1}\left|f_j^{(t)}(\boldsymbol{x}(s)) - \tilde{f}_j^{(t)}(s)\right| \leq O\left(\frac{\log^{L+1} m}{\sqrt{c}}\right)\|\boldsymbol{v}\|_2. \tag{6}$$

### 3.1 INTUITION BEHIND IMPLICIT REGULARIZATION

To prove Theorem 2, we introduce our key analysis that the gradient of $g^{l,(t)}(\boldsymbol{x}(s))$ is "Gaussian random walk" and nearly equal to a straight line on the linear path $\boldsymbol{x}(s)$, which we call **random walk analysis**. For simplicity, we explain the outline of the proof for the initial state of the network (i.e. $t = 0$). All proofs are given in the Supplementary Material (including $0 \leq t \leq T$).

To state our key Lemma, we define the following function: $\breve{g}^l(s) := \mathbf{W}^l\phi(\tilde{g}^{l-1}(s))$, which is an analogue of the unit output $g^l(\boldsymbol{x}(s)) = \mathbf{W}^l\phi(g^{l-1}(\boldsymbol{x}(s)))$. In other words, $\breve{g}^l(s)$ is the one in which $g^{l-1}(\boldsymbol{x}(s))$ is replaced by $\tilde{g}^{l-1}(s)$. Next, Lemma 3 shows that each $\breve{g}_i^l$ can be well approximated by a linear function $\tilde{g}_i^l(s)$ with high probability.

**Lemma 3** (Linear Approximation Analogue). *With probability at least $1 - e^{-\Omega(\log^2 m)}$, for every $\boldsymbol{x}^{(p)}$, $\boldsymbol{x}^{(q)}$ $(p,q \in [n]$, $p \neq q)$, $t \in [T]$ and $l \in [L]$, we have*

$$\sup_{0 \leq s \leq 1}\left|\breve{g}_i^{l,(t)}(s) - \tilde{g}_i^{l,(t)}(s)\right| \leq O\left(\frac{\log^2 m}{\sqrt{m}}\right)\|\boldsymbol{v}\|_2, \quad (\forall i \in [m]), \tag{7}$$

$$\sup_{0 \leq s \leq 1}\left|\breve{f}_j^{(t)}(s) - \tilde{f}_j^{(t)}(s)\right| \leq O\left(\frac{\log^2 m}{\sqrt{c}}\right)\|\boldsymbol{v}\|_2, \quad (\forall j \in [c]). \tag{8}$$

The purpose of this subsection is to give an intuitive explanation of the proof of Lemma 3, which can be divided into two steps. The first step gives estimates of the number of **breakpoints**. We show that the number of breakpoints of the piecewise linear function $\breve{g}_i^l(s)$ is less than or equal to the number of units in the layer (i.e. $m$).

The second step gives estimates of **gradient gaps** of $\breve{g}_i^l(s)$. We show that the gradient gaps are independent Gaussian random variables and the gradient of $\breve{g}^l(s)$ is the sum of the gradient gaps, which indicates that the gradient is a "Gaussian random walk". Note that in this setting, the number of breakpoints is equal to the total number of steps of the random walk.

---

[4] Although the components of $g^l(s)$, $f(s)$ are non-differentiable at breakpoints, with probability 1, the start point $s = 0$ is not a breakpoint.

**Step 1.** By the definition of $\breve{g}^l(s)$, we may write

$$\breve{g}_i^l(s) = \sum_{j=1}^m [\mathbf{W}^l]_{i,j} \, \phi(\tilde{g}_j^{l-1}(s)) \, . \tag{9}$$

Note that the input to the above $\phi$, that is $\tilde{g}_j^{l-1}(s)$, is a linear function on $s \in [0,1]$. For each $j \in [m]$, the linear equation $\tilde{g}_j^{l-1}(s) = 0$ has at most one solution ($s = s^*$). If the solution $s = s^*$ satisfies $0 < s^* < 1$, then $\breve{g}_i^l(s)$ has a breakpoint at $s = s^*$. This shows that the number of breakpoints of $\breve{g}_i^l(s)$ is bounded by the number of the linear equations $\{\tilde{g}_j^{l-1}(s) = 0\}_{j \in [m]}$, which is equal to $m$. We denote by $0 < s_1 < s_2 < \cdots < s_\alpha < 1$ breakpoints of $\breve{g}_i^l(s)$, where $\alpha$ is the number of breakpoints. For $\beta \in [\alpha]$, we set an open interval of breakpoints $\mathbb{I}_\beta := (s_\beta, s_{\beta+1})$.

**Step 2.** To give an intuition that the gradient of $\breve{g}_i^l(s) - \tilde{g}_i^l(s)$ is "Gaussian random walk", we fix some notation. For $\beta \in [\alpha]$, we define the gradient $\nabla_\beta$:

$$\nabla_\beta := \frac{d}{ds} \left( \breve{g}_i^l(s) - \tilde{g}_i^l(s) \right) \Big|_{s \in \mathbb{I}_\beta} . \tag{10}$$

Note that $\tilde{g}_i^l(s)$ is a linear function, and there is no breakpoints for $\breve{g}_i^l(s)$ on $\mathbb{I}_\beta$. Note also that since $\breve{g}_i^l(s) - \tilde{g}_i^l(s)$ is linear on $\mathbb{I}_\beta$, the gradient is constant on $\mathbb{I}_\beta$. For $\beta \in [\alpha]$, we define a **gradient gap**: $\mathsf{x}_\beta := \nabla_\beta - \nabla_{\beta-1}$, and we have the following estimate with probability at least $1 - e^{-\Omega(\log^2 m)}$:

$$\mathsf{x}_\beta = S\omega_\beta \quad \text{where} \quad \omega_\beta \sim \mathcal{N}(0, 2/m), \quad |S| \leq \frac{\log m}{\sqrt{m}} \|\boldsymbol{v}\|_2. \tag{11}$$

Note that $\omega_\beta$ is an element of the weight matrix $\mathbf{W}^l$. Thus, $\{\mathsf{x}_\beta\}$ are independent Gaussian random variables, and $\nabla_\beta$ is the sum of $\mathsf{x}_\beta$, that is

$$\nabla_\beta = \sum_{\gamma=1}^\beta \mathsf{x}_\gamma, \quad (\beta \in [\alpha]). \tag{12}$$

This shows that the gradient $\nabla_\beta$ is a "Gaussian random walk" and depends essentially on the number of breakpoints $\alpha$ and the variance of weights $2/m$. Using the randomness of $\mathbf{W}^l$ and general Hoeffding's inequality (Vershynin, 2018), we have

$$\mathbb{P}\left[ \Big| \sum_{\gamma=1}^\beta S\omega_\gamma \Big| \leq \varepsilon \right] \geq 1 - e^{-\Omega\left( \frac{m^2 \varepsilon^2}{\alpha \|v\|_2^2 \log^2 m} \right)}. \tag{13}$$

According to Step 1, the number of breakpoints $\alpha$ is less than or equal to $m$. Thus, $\nabla_\beta$ is bounded by $\frac{\log^2 m}{\sqrt{m}} \|\boldsymbol{v}\|_2$ with probability at least $1 - e^{-\Omega(\log^2 m)}$. Therefore, with the same probability, we have $\left| \frac{d}{ds} \left( \breve{g}_i^l(s) - \tilde{g}_i^l(s) \right) \right| \leq \frac{\log m}{\sqrt{m}} \|\boldsymbol{v}\|_2, \ (\forall s \in [0,1])$. Thus, integrating this inequality from $s = 0$ to $s = 1$ shows the estimates of (7) in Lemma 3. Note that the proof of (8) in Lemma 3 is idential to the above proof, except the fact that each entry of $\mathbf{B}$ follows from $\mathcal{N}(0, 2/c)$ insted of $\mathcal{N}(0, 2/m)$.

Note that as illustrated in Figure 1, for sufficiently large $m$, the gradient $\nabla_\beta$ can be made arbitrarily small, which means that the networks interpolate almost linearly between the data points.

## 3.2 Proof Theorem 2

The purpose of this subsection is to give the proof of Theorem 2. We proceed by induction on the layer $l$. Note that since $g^0(\boldsymbol{x}(s)) = \mathbf{A}\boldsymbol{x}(s)$ is linear and $\tilde{g}^0(s) = g^0(\boldsymbol{x}(s))$, $\breve{g}^1(s)$ is identically equal to $g^1(\boldsymbol{x}(s)) = \mathbf{W}^1 \phi(A\boldsymbol{x}(s))$. Thus, the case $l = 1$ is true. Assume the theorem holds for any layer $l = k$, and let us prove it for $l = k + 1$. Using the triangle inequality, we have

$$\left| g_i^{k+1}(\boldsymbol{x}(s)) - \tilde{g}_i^{k+1}(s) \right| \leq \left| g_i^{k+1}(\boldsymbol{x}(s)) - \breve{g}_i^{k+1}(s) \right| + \left| \breve{g}_i^{k+1}(s) - \tilde{g}_i^{k+1}(s) \right|. \tag{14}$$

By Proposition 11.3 of (Allen-Zhu et al., 2018b), for any $y, \tilde{y} \in \mathbb{R}^m$, there exists a diagonal matrix $\boldsymbol{D} \in \mathbb{R}^{m \times m}$ such that $|D_{j,j}| \leq 1$ and $\phi(\boldsymbol{y}) - \phi(\tilde{\boldsymbol{y}}) = \boldsymbol{D}(\boldsymbol{y} - \tilde{\boldsymbol{y}})$. Thus, for the first term on the RHS, we may write

$$\left| g_i^{k+1}(\boldsymbol{x}(s)) - \breve{g}_i^{k+1}(s) \right| = \left| \left[ \mathbf{W}^{k+1} \phi\big(g^k(\boldsymbol{x}(s))\big) \right]_i - \left[ \mathbf{W}^{k+1} \phi(\tilde{g}^k(s)) \right]_i \right| \tag{15}$$

$$= \left| \left[ \mathbf{W}^{k+1} \boldsymbol{D}\big(g^k(\boldsymbol{x}(s)) - \tilde{g}^k(s)\big) \right]_i \right| = \left| \sum_{\rho=1}^m [\mathbf{W}^k]_{i,\rho} D_{\rho,\rho}\big(g_\rho^k(\boldsymbol{x}(s)) - \tilde{g}_\rho^k(s)\big) \right| \tag{16}$$

$$\leq \left\| g^k(\boldsymbol{x}(s)) - \tilde{g}^k(s) \right\|_2 \frac{\log m}{\sqrt{m}}. \tag{17}$$

In the last inequality, we use general Hoeffding's inequality (Vershynin, 2018), and this inequality holds with probability at least $1 - e^{-\Omega(\log^2 m)}$. We can now apply our induction hypothesis to the above estimate. Thus, applying (7) in Lemma 3 to the second term on the RHS of (14) shows the estimates of (5) in Theorem 2. Next, Using the triangle inequality for $f_j^{(t)}$, we have

$$\left| f_j^{(t)}(\boldsymbol{x}(s)) - \tilde{f}_j^{(t)}(s) \right| \leq \left| f_j^{(t)}(\boldsymbol{x}(s)) - \breve{f}_j^{(t)}(s) \right| + \left| \breve{f}_j^{(t)}(s) - \tilde{f}_j^{(t)}(s) \right|. \tag{18}$$

In a similar way, using the randomness of $\mathbf{B}$ (recall each entry of $\mathbf{B}$ follows from $\mathcal{N}(0, 2/c)$), with probability at least $1 - e^{-\Omega(\log^2 m)}$, we have

$$\left| f_j^{(t)}(\boldsymbol{x}(s)) - \breve{f}_j^{(t)}(s) \right| \leq \left\| g^L(\boldsymbol{x}(s)) - \tilde{g}^L(s) \right\|_2 \sqrt{\frac{m}{c}} \log m \leq O\left( \frac{\log^3 m}{\sqrt{c}} \right) \|\boldsymbol{v}\|_2. \tag{19}$$

In the last inequality, we use (7) in Lemma 3. Thus, applying (8) in Lemma 3 to the second term on the RHS of (18) shows the estimates of (6) in Theorem 2. We complete the proof.

## 4 GENERALIZATION

In this section, considering the mechanism of implicit regularization revealed by random walk analysis, we provide *a priori* estimates for the generalization performance of over-parameterized deep neural networks, in an $l^2$ regression task on a one-dimensional input-space. Since the network is over-parameterized, the expressive power of the network is rich enough to considerably overfit the data. Nevertheless, it is known empirically that properly initialized over-parameterized deep neural networks can achieve the good generalization performance while fitting all training data. We estimate the low complexity of over-parameterized deep networks and show that the error between a trained over-parameterized neural network and the target function can be uniformly bounded by an arbitrarily small positive number.

**Setting.** We propose a new type of generalization bounds (Theorem 4), and prove this theorem in models with one dimensional input: $d = 1$. In the following, we restrict ourselves to a one-dimensional regression task on an interval $[0, \nu] \subset \mathbb{R}$. The training dataset $\{(x^{(i)}, \boldsymbol{y}^{(i)})\}_{i \in [n]}$ is given as $n$ i.i.d. samples from some unknown distribution $\mathcal{D}$. We assume that the corresponding target function for the regression task $f^* : [0, \nu] \to \mathbb{R}^c$ is a $C^1$-class function. This implies that $\boldsymbol{y}^{(i)} = f^*(x^{(i)})$. Without loss of generality, we may assume that the input data $\{x^{(i)}\}_{i \in [n]}$ follow a uniform distribution on $[0, \nu]$, and after relabeling, we may assume that the data points $\{x^{(1)}, \ldots, x^{(n)}\}$ are ordered by index: $0 < x^{(1)} < \cdots < x^{(n)} < \nu$. We define $(x^{(0)}, y^{(0)}) := (0, f^*(x^{(0)}))$, $(x^{(n+1)}, y^{(n+1)}) := (\nu, f^*(\nu))$ and $\mathcal{T} := \sup_{k \in [n+1]} (x^{(k)} - x^{(k-1)})$. We denote by $\hat{f} : [0, \nu] \to \mathbb{R}$ the linear interpolation of the data points $\{(x^{(i)}, f^*(x^{(i)}))\}_{i \in [n]}$.

**A priori generalization bounds.** Now, we introduce a novel approach for *a priori* generalization bounds, which is based on random walk analysis in the over-parameterized regime. This can be interpreted as the significant expressive power of over-parameterized neural networks is controlled by implicit regularization.

**Theorem 4.** *Suppose $f^*(x)$ is a $C^1$- class function on $[0, \nu]$. Under the same setting as Theorem 1, for $\delta \in (0, 1/2]$, then with probability at least $1 - (\delta + e^{-\Omega(\log^2 m)})$, we have*

$$\mathbb{E}_{(x,y)\sim\mathcal{D}}\left[\ell\left(f^{(T)}(x), \boldsymbol{y}\right)\right] \leq O\left(\frac{\nu^2 c}{n^2 \delta^2}\right). \tag{20}$$

**Proof sketch of generalization.** The purpose of this paragraph is to give an intuitive explanation of the proof of Theorem 4 for one-dimensional output $c = 1$. All proofs are given in the Supplementary Material.

To estimate $\mathbb{E}_{(x,y)\sim\mathcal{D}}\left[\ell\left(f^{(T)}(x), y\right)\right]$, we evaluate $\left\|f^{(T)}(x) - f^*(x)\right\|_2$, which we may write as

$$\left\|f^{(T)}(x) - f^*(x)\right\|_2 \leq \left|f^{(T)}(x) - \hat{f}(x)\right| + \left|\hat{f}(x) - f^*(x)\right| \tag{21}$$

The first term of the RHS of (21) represents an error between the trained network output $f^{(T)}(x)$ and the piecewise linear function $\hat{f}(x)$. We can use the results of **random walk analysis** to evaluate this critical term as follows. Theorem 2 provides that the linear approximation error of the network output, which we denote by $\varepsilon_{f^{(T)}}$, is small in each interval $[x^{(k-1)}, x^{(k)}]$, and hence in each interval $[x^{(k-1)}, x^{(k)}]$, the difference between $f^{(T)}(x)$ and $\hat{f}(x)$ falls within the error $\varepsilon_{f^{(T)}}$. This suggests that the network output between adjacent training points is properly controlled by weight initialization and SGD to keep connecting the points almost straight, which results in low complexity of over-parameterized neural networks. This statement can be extended to the interval $[0, \nu]$. Fix $\varepsilon > 0$, and suppose $m \geq \widetilde{O}\left(\mathcal{T}^6/\varepsilon^3\right)$. Then with high probability at least $1 - e^{-\Omega(\log^2 m)}$, we have

$$\sup_{x\in[0,\nu]}\left|f^{(T)}(x) - \hat{f}(x)\right| \leq \sqrt{\frac{\varepsilon}{2}} \tag{22}$$

The second term of RHS of (21) is the error of the piecewise linear approximation of $f^*(x)$ by $\hat{f}(x)$ on the interval $[0, \nu]$. Note that the error can be reduced by increasing $n$. Hence, for any fixed $\varepsilon > 0$, there exists $\delta > 0$ such that if $n \geq O\left(\frac{\nu}{\sqrt{\varepsilon}\delta}\right)$, with probability at least $1 - \delta$, we have $\sup_{x\in[0,\nu]}|\hat{f}(x) - f^*(x)| \leq \sqrt{\frac{\varepsilon}{2}}$. Thus, from (21), for $m$ and $n$ sufficiently large, the error between the trained network output and the target function is uniformly bounded by $\varepsilon$ on the interval $[0, \nu]$ with probability at least $1 - \left(e^{-\Omega(\log^2 m)} + \delta\right)$. This gives *a priori* estimates for the generalization performance of over-parameterized neural networks.

## 5 RELATED WORK

Implicit regularization in neural networks has recently become an active area of research in machine learning. A number of works have focused on the behavior of gradient descent on over-parameterized neural networks (Neyshabur et al., 2014; Lin et al., 2016; Zhang et al., 2016; Soudry et al., 2018; Rahaman et al., 2018). In order to get a handle on implicit regularization in deep neural networks, the majority of theoretical attention has been devoted to linear neural networks (Ji & Telgarsky, 2018; Gidel et al., 2019; Arora et al., 2019a).

Many works try to explain generalization of over-parameterized neural networks. Recent works have shown that on sufficiently over-parameterized neural networks, the learning dynamics of gradient descent are governed by the NTK (Daniely et al., 2016; Jacot et al., 2018; Du et al., 2018; Allen-Zhu et al., 2018b; Lee et al., 2019; Arora et al., 2019b). In these settings, the implicit regularization and the generalization error of the resulting network can be analyzed via NTK and the reproducing kernel Hilbert space (RKHS) (Bietti & Mairal, 2019; Nakkiran et al., 2019). To extend it to SGD, Hayou et al. (2019) introduce a stochastic differential equation dependent on the NTK. Closely related work Cao & Gu (2019) showed that the expected 0-1 loss of a wide enough ReLU network trained with SGD and random initialization can be bounded by the training loss of a random feature model induced by the network gradient at initialization.

This contrasts with other recent results that show a provable separation between the generalization error obtained by neural networks and kernel methods (Wei et al., 2018; Allen-Zhu et al., 2018a;

Allen-Zhu & Li, 2019). Several papers suggest that training deep models with gradient descent can behave differently from kernel methods, and have much richer implicit regularization (Chizat et al., 2019; Woodworth et al., 2019; Yehudai & Shamir, 2019; Geiger et al., 2019).

# 6 CONCLUSION

In this work, probability estimates for the network output behavior (i.e. random walk analysis) provide *a priori* generalization estimates for $l^2$ regression problems. We prove that even after training, network gradients between the data points are approximately Gaussian random walks, and the variation of the gradients between the data points is extremely small and depends essentially on the number of breakpoints and the variance of weights. To the best of our knowledge, this paper is the first to show a mechanism of implicit regularization and to prove the generalization bounds by using the implicit regularization for deep (three or more layer) neural networks with ReLU activation. As a result, we also show that over-parameterized deep neural networks can learn $C^1$-class functions. Importantly, our analysis is independent of the kernel generalization analysis, and the generalization bounds are different from the NTK inductive bias of the RKHS norm.

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

# A    PROOFS

## A.1    PROOF OF LEMMA 3

**Lemma 3** (reshown, see §3.1). *With probability at least* $1 - e^{-\Omega(\log^2 m)}$, *for every* $\boldsymbol{x}^{(p)}$, $\boldsymbol{x}^{(q)}$ $(p, q \in [n], p \neq q)$, $t \in [T]$ *and* $l \in [L]$, *we have*

$$\sup_{0 \leq s \leq 1} \left| \breve{g}_i^{l,(t)}(s) - \tilde{g}_i^{l,(t)}(s) \right| \leq O\left( \frac{\log^2 m}{\sqrt{m}} \right) \|\boldsymbol{v}\|_2, \quad (\forall i \in [m]), \tag{23}$$

$$\sup_{0 \leq s \leq 1} \left| \breve{f}_j^{(t)}(s) - \tilde{f}_j^{(t)}(s) \right| \leq O\left( \frac{\log^2 m}{\sqrt{c}} \right) \|\boldsymbol{v}\|_2, \quad (\forall j \in [c]). \tag{24}$$

*Proof.* We prove Lemma 3 for a fixed $l \in [L]$ and $t \in [T]$ every pair of data points $\boldsymbol{x}^{(p)}$, $\boldsymbol{x}^{(q)}$ $(p, q \in [n], p \neq q)$, because we can apply union bound at the end.

Recall the definition $g^{l,(t)}(x)$ (the unit output), $\mathbf{G}^{l,(t)}(x)$ (the indicator matrix function) and $\boldsymbol{x}(s) = (1-s)\boldsymbol{x}^{(p)} + s\boldsymbol{x}^{(q)}$, $s \in [0, 1]$ (a linearly interpolating path between data points $\boldsymbol{x}^{(p)}$, $\boldsymbol{x}^{(q)}$), we have

$$g^{l,(t)}(\boldsymbol{x}(s)) = \mathbf{W}^{l,(t)} \mathbf{G}^{l-1,(t)}(\boldsymbol{x}(s)) \mathbf{W}^{l-1,(t)} \cdots \mathbf{W}^{1,(t)} \mathbf{G}^{0,(t)}(\boldsymbol{x}(s)) \mathbf{A} \boldsymbol{x}(s). \tag{25}$$

Using the definition of $\tilde{g}^{l,(t)}(s)$ and $\breve{g}^{l,(t)}(s)$, we also have

$$\tilde{g}^{l,(t)}(s) = \mathbf{W}^{l,(t)} \mathbf{G}^{l-1,(t)}(\boldsymbol{x}(0)) \mathbf{W}^{l-1,(t)} \cdots \mathbf{W}^{1,(t)} \mathbf{G}^{0,(t)}(\boldsymbol{x}(0)) \mathbf{A} \boldsymbol{x}(s), \tag{26}$$

$$\tag{27}$$

$$\breve{g}^{l,(t)}(s) = \mathbf{W}^{l,(t)} \phi(\tilde{g}^{l-1,(t)}(s)) \tag{28}$$

$$= \mathbf{W}^{l,(t)} \widetilde{\mathbf{G}}^{l-1,(t)}(s) \mathbf{W}^{l-1,(t)} \mathbf{G}^{l-2,(t)}(\boldsymbol{x}(0)) \mathbf{W}^{l-2,(t)} \cdots \mathbf{W}^{1,(t)} \mathbf{G}^{0,(t)}(\boldsymbol{x}(0)) \mathbf{A} \boldsymbol{x}(s), \tag{29}$$

where $\widetilde{\mathbf{G}}^{l-1,(t)}(s)$ is an indicator diagonal matrix for $s \in [0, 1]$ as follows:

$$[\widetilde{\mathbf{G}}^{l-1,(t)}(s)]_{i,i} := \mathbf{1}_{\left\{ \tilde{g}_i^{l-1,(t)}(s) \geq 0 \right\}}, \quad [\widetilde{\mathbf{G}}^{l-1,(t)}(s)]_{i,j} := 0 \ (i \neq j). \tag{30}$$

We begin by proving that the number of breakpoints of the function $\breve{g}_i^{l,(t)}(s)$ in the interval $[0.1]$ is bounded by $m$. For fixed $i \in [m]$, we have

$$\breve{g}_i^{l,(t)}(s) = [\mathbf{W}^{l,(t)} \phi(\tilde{g}^{l-1,(t)}(s))]_i \tag{31}$$

$$= \sum_{j=1}^m [\mathbf{W}^{l,(t)}]_{i,j} \, \phi(\tilde{g}_j^{l-1,(t)}(s)) \tag{32}$$

$$= \sum_{j=1}^m [\mathbf{W}^{l,(t)}]_{i,j} \, \tilde{g}_j^{l-1,(t)}(s) \mathbf{1}_{\left\{ \tilde{g}_j^{l-1,(t)}(s) \geq 0 \right\}} \tag{33}$$

Note that the input to the above $\phi$, that is $\tilde{g}_j^{l-1,(t)}(s)$, is a linear function on $s \in [0, 1]$. For each $j \in [m]$, the linear equation $\tilde{g}_j^{l-1,(t)}(s) = 0$ has at most one solution $s = s^*$.[5] If the solution $s = s^*$ satisfies inequality $0 < s^* < 1$, then $\breve{g}_i^{l,(t)}(s)$ has a breakpoint at $s = s^*$. This shows that the number of breakpoints of $\breve{g}_i^{l,(t)}(s)$ is bounded by the number of the linear equations

$$\tilde{g}_j^{l-1,(t)}(s) = 0 \ (j \in [m]), \tag{34}$$

which is clearly equal to $m$.

We denote by $0 < s_1 < s_2 < \cdots < s_\alpha < 1$ all breakpoints of $\breve{g}_i^{l,(t)}(s)$, where $\alpha$ is the number of breakpoints. For notational simplicity, we set $s_0 = 0$ and $s_{\alpha+1} = 1$. Note that, with probability 1,

---

[5]Otherwise, the linear function is identically zero (i.e. $\tilde{g}_j^{l-1,(t)}(s) \equiv 0$), which does not affect the number of breakpoints.

breakpoints are all distinct from each other. Each breakpoint $s_\beta$ $(\beta \in [\alpha])$ corresponds to a linear equation $\widetilde{g}_j^{\,l-1,(t)}(s) = 0$ for some $j \in [m]$. In other words, for each breakpoint $s_\beta$ $(\beta \in [\alpha])$, there exists a unique element $j \in [m]$ such that $\widetilde{g}_j^{\,l-1,(t)}(s) = 0$. Therefore, for $s = s_\beta$, we set $j = j_\beta$ $(\beta \in [\alpha])$ then we have $\widetilde{g}_{j_\beta}^{\,l-1,(t)}(s_\beta) = 0$.

It is easy to verify $\frac{d}{ds}\boldsymbol{x}(s) = \boldsymbol{x}^{(q)} - \boldsymbol{x}^{(p)} =: \boldsymbol{v}$. Therefore,

$$\frac{d}{ds}\widetilde{g}^{\,l,(t)}(s) = \mathbf{W}^{l,(t)}\mathbf{G}^{l-1,(t)}(\boldsymbol{x}(0))\mathbf{W}^{l-1,(t)}\mathbf{G}^{l-2,(t)}(\boldsymbol{x}(0))\mathbf{W}^{l-2,(t)}\cdots\mathbf{W}^{1,(t)}\mathbf{G}^{0,(t)}(\boldsymbol{x}(0))\mathbf{A}\boldsymbol{v}, \tag{35}$$

$$\frac{d}{ds}\breve{g}^{\,l,(t)}(s) = \mathbf{W}^{l,(t)}\widetilde{\mathbf{G}}^{l-1,(t)}(s)\mathbf{W}^{l-1,(t)}\mathbf{G}^{l-2,(t)}(\boldsymbol{x}(0))\mathbf{W}^{l-2,(t)}\cdots\mathbf{W}^{1,(t)}\mathbf{G}^{0,(t)}(\boldsymbol{x}(0))\mathbf{A}\boldsymbol{v}. \tag{36}$$

This implies,

$$\frac{d}{ds}\breve{g}^{\,l,(t)}(s) - \frac{d}{ds}\widetilde{g}^{\,l,(t)}(s) \tag{37}$$

$$= \mathbf{W}^{l,(t)}\left(\widetilde{\mathbf{G}}^{l-1,(t)}(s) - \mathbf{G}^{l-1,(t)}(\boldsymbol{x}(0))\right)\mathbf{W}^{l-1,(t)}\mathbf{G}^{l-2,(t)}(\boldsymbol{x}(0))\cdots\mathbf{W}^{1,(t)}\mathbf{G}^{0,(t)}(\boldsymbol{x}(0))\mathbf{A}\boldsymbol{v}. \tag{38}$$

Note that from the definition of the indicator matrix, $[\mathbf{G}^{l-1,(t)}(\boldsymbol{x}(0))]_{i,i} = \mathbf{1}_{\left\{[g^{l-1,(t)}(\boldsymbol{x}(0))]_i \geq 0\right\}}$, and from the definition of the linear approximation, $\widetilde{g}^{\,l-1,(t)}(0) = g^{l-1,(t)}(\boldsymbol{x}(0))$, which says that $\widetilde{\mathbf{G}}^{l-1,(t)}(0)$ equals $\mathbf{G}^{l-1,(t)}(\boldsymbol{x}(0))$.

For each $i \in [m]$, we may write

$$\frac{d}{ds}\breve{g}_i^{\,l,(t)}(s) - \frac{d}{ds}\widetilde{g}_i^{\,l,(t)}(s) = \sum_{j=1}^m [\mathbf{W}^{l,(t)}]_{i,j}[\mathbf{M}^{(t)}\boldsymbol{v}]_j\widetilde{\lambda}_j, \tag{39}$$

where $\mathbf{M}^{(t)}$ is a matrix:

$$\mathbf{M}^{(t)} := \mathbf{W}^{l-1,(t)}\mathbf{G}^{l-2,(t)}(\boldsymbol{x}(0))\mathbf{W}^{l-2,(t)}\cdots\mathbf{W}^{1,(t)}\mathbf{G}^{0,(t)}(\boldsymbol{x}(0))\mathbf{A} \tag{40}$$

and $\widetilde{\lambda}_j$ is the difference of indicator functions:

$$\widetilde{\lambda}_j := \mathbf{1}_{\left\{\widetilde{g}_j^{\,l-1,(t)}(s) \geq 0\right\}} - \mathbf{1}_{\left\{\widetilde{g}_j^{\,l-1,(t)}(0) \geq 0\right\}}. \tag{41}$$

For $\beta \in [\alpha]$, we set an open interval of breakpoints $\mathbb{I}_\beta := (s_\beta, s_{\beta+1})$, and recall the notation of the gradient $\nabla_\beta$

$$\nabla_\beta := \left.\frac{d}{ds}\left(\breve{g}_i^{l,(t)}(s) - \widetilde{g}_i^{l,(t)}(s)\right)\right|_{s \in \mathbb{I}_\beta}, \tag{42}$$

and the gradient gap $\mathsf{x}_\beta := \nabla_\beta - \nabla_{\beta-1}$. Note that since there is not a breakpoint in $\mathbb{I}_\beta$, the gradient $\nabla_\beta$ is constant in $\mathbb{I}_\beta$.

Hence, for $\beta \in [\alpha]$ and $s \in \mathbb{I}_\beta$, we may write

$$\frac{d}{ds}\breve{g}_i^{l,(t)}(s) - \frac{d}{ds}\widetilde{g}_i^{l,(t)}(s) = \sum_{\gamma=1}^\beta \mathsf{x}_\gamma. \tag{43}$$

Note that the gradient gap $\mathsf{x}_\gamma$ $(\gamma \in [\alpha])$ can be rewritten as

$$\mathsf{x}_\gamma = [\mathbf{W}^{l,(t)}]_{ij_\gamma}[\mathbf{M}^{(t)}v]_{j_\gamma}\widetilde{\zeta}_\gamma, \tag{44}$$

where $\widetilde{\zeta}_\gamma$ is a difference of indicator functions:

$$\widetilde{\zeta}_\gamma := \mathbf{1}_{\left\{\widetilde{g}_{j_\gamma}^{\,l-1,(t)}(d_\gamma) \geq 0\right\}} - \mathbf{1}_{\left\{\widetilde{g}_{j_\gamma}^{\,l-1,(t)}(d_{\gamma-1}) \geq 0\right\}}, \qquad \forall\, d_{\gamma-1} \in \mathbb{I}_{\gamma-1} \quad \text{and} \quad \forall\, d_\gamma \in \mathbb{I}_\gamma. \tag{45}$$

Note that $\widetilde{\zeta}_\gamma$ is independent of the choice of $d_{\gamma-1}$ and $d_\gamma$. Note also that since the linear function $\widetilde{g}_{j_\gamma}^{l-1,(t)}(s)$ switches the sign at $s = s_\gamma$, we have $\widetilde{\zeta}_\gamma = \pm 1$.

We begin by proving the upper bound on $\sum_{\gamma=1}^\beta x_\gamma$. For notational simplicity, we use $\mathbf{W}^l$ to denote $\mathbf{W}^{l,(0)}$, and for $l \in [L]$ we set

$$\mathbf{dW}^l := \mathbf{W}^{l,(t)} - \mathbf{W}^{l,(0)} = \mathbf{W}^{l,(t)} - \mathbf{W}^l, \tag{46}$$

then we may write for $\gamma \in [\alpha]$,

$$[\mathbf{M}^{(t)}\boldsymbol{v}]_{j_\gamma} = [\mathbf{W}^{l-1,(t)}\mathbf{G}^{l-2,(t)}(\boldsymbol{x}(0))\mathbf{W}^{l-2,(t)}\cdots\mathbf{W}^{1,(t)}\mathbf{G}^{0,(t)}(\boldsymbol{x}(0))\mathbf{A}\boldsymbol{v}]_{j_\gamma} \tag{47}$$

$$= \sum_{\rho=1}^m [\mathbf{W}^{l-1,(t)}]_{j_\gamma,\rho}[\mathbf{N}\boldsymbol{v}]_\rho \tag{48}$$

$$= \sum_{\rho=1}^m [\mathbf{W}^{l-1}]_{j_\gamma,\rho}[\mathbf{N}\boldsymbol{v}]_\rho + \sum_{\rho=1}^m [\mathbf{dW}^{l-1}]_{j_\gamma,\rho}[\mathbf{N}\boldsymbol{v}]_\rho \tag{49}$$

$$=: \mathrm{h}_\gamma + \mathrm{dh}_\gamma, \tag{50}$$

where $\mathbf{N} := \mathbf{G}^{l-2,(t)}(\boldsymbol{x}(0))\mathbf{W}^{l-2,(t)}\cdots\mathbf{W}^{1,(t)}\mathbf{G}^{0,(t)}(\boldsymbol{x}(0))\mathbf{A}$.

Therefore, we have

$$\mathrm{x}_\gamma = \widetilde{\zeta}_\gamma[\mathbf{W}^{l,(t)}]_{i,j_\gamma}[\mathbf{M}^{(t)}\boldsymbol{v}]_{j_\gamma} \tag{51}$$

$$= \widetilde{\zeta}_\gamma\left([\mathbf{W}^l]_{i,j_\gamma} + [\mathbf{dW}^l]_{i,j_\gamma}\right)\left(\mathrm{h}_\gamma + \mathrm{dh}_\gamma\right) \tag{52}$$

$$= \widetilde{\zeta}_\gamma[\mathbf{W}^l]_{i,j_\gamma}\mathrm{h}_\gamma + \widetilde{\zeta}_\gamma[\mathbf{dW}^l]_{i,j_\gamma}\mathrm{h}_\gamma + \widetilde{\zeta}_\gamma[\mathbf{W}^l]_{i,j_\gamma}\mathrm{dh}_\gamma + \widetilde{\zeta}_\gamma[\mathbf{dW}^l]_{i,j_\gamma}\mathrm{dh}_\gamma \tag{53}$$

$$=: \mathrm{x}_\gamma^{(0)} + \mathrm{x}_\gamma^{(1)} + \mathrm{x}_\gamma^{(2)} + \mathrm{x}_\gamma^{(3)}. \tag{54}$$

In order to estimate $\mathbf{N}\boldsymbol{v}$, we use the following estimate (see Lemma 7.1 and Claim 11.2 in (Allen-Zhu et al., 2018b)):

**Lemma.** *If $\varepsilon \in (0,1]$, with probability at least $1 - e^{-\Omega(m\varepsilon^2/L)}$, for a fixed vector $\boldsymbol{z}$ and $l \in [L]$, we have*

$$\|\mathbf{G}^{l,(t)}(\boldsymbol{x}(0))\mathbf{W}^{l,(t)}\cdots\mathbf{W}^{1,(t)}\mathbf{G}^{0,(t)}(\boldsymbol{x}(0))\mathbf{A}\boldsymbol{z}\|_2 \leq (1+\varepsilon)\|\boldsymbol{z}\|_2. \tag{55}$$

This Lemma implies that with probability at least $1 - e^{-\Omega(m/L)}$, we have

$$\|\mathbf{N}\boldsymbol{v}\|_2 = \|\mathbf{G}^{l-2,(t)}(\boldsymbol{x}(0))\mathbf{W}^{l-2,(t)}\cdots\mathbf{W}^{1,(t)}\mathbf{G}^{0,(t)}(\boldsymbol{x}(0))\mathbf{A}\boldsymbol{v}\|_2 \leq 2\|\boldsymbol{v}\|_2. \tag{56}$$

We will estimate the 1st term $\mathrm{x}_\gamma^{(0)}$. Conditioning on this event (56) happens, using the randomness of $\mathbf{W}^{l-1}$ and general Hoeffding's inequality (Vershynin, 2018), for each fixed vector $\boldsymbol{v}$, we have

$$\mathbb{P}\left\{|\mathrm{h}_\gamma| > \tau\right\} \leq 2\exp\left(-\frac{C\tau^2}{(2/m)\|\mathbf{N}\boldsymbol{v}\|_2^2}\right). \tag{57}$$

Choose $\tau := \frac{\log m}{\sqrt{m}}\|\mathbf{N}\boldsymbol{v}\|_2$. Then, with probability at least $1 - e^{-\Omega(\log^2 m)}$, we have

$$|\mathrm{h}_\gamma| \leq \frac{\log m}{\sqrt{m}}\|\mathbf{N}\boldsymbol{v}\|_2. \tag{58}$$

We set $\mathrm{u}_\gamma^{(0)} := \widetilde{\zeta}_\gamma\mathrm{h}_\gamma$ and $\mathbf{u}^{(0)} = (\mathrm{u}_1^{(0)},\ldots,\mathrm{u}_\beta^{(0)})$. Note that since $\widetilde{\zeta}_\gamma = \pm 1$, we have $\left|\mathrm{u}_\gamma^{(0)}\right| = \left|\widetilde{\zeta}_\gamma\mathrm{h}_\gamma\right| \leq \frac{\log m}{\sqrt{m}}\|\mathbf{N}\boldsymbol{v}\|_2$. The set of the gradient gaps $\{\mathrm{x}_\gamma^{(0)}\} = \{\mathrm{u}_\gamma^{(0)}[\mathbf{W}^l]_{i,j_\gamma}\}$ are a sequence of independent Gaussian random variables. Hence, $\sum_{\gamma=1}^\beta \mathrm{x}_\gamma^{(0)}$ is a "Gaussian random walk" (12). Thus, using the randomness of $\mathbf{W}^l$, conditioning on the above event (58), fixing any $\boldsymbol{v}$, with probability at

least $1 - e^{-\Omega(\log^2 m)}$, we have

$$\left| \sum_{\gamma=1}^{\beta} \mathrm{x}_\gamma^{(0)} \right| \leq \frac{\log m}{\sqrt{m}} \|\mathbf{u}^{(0)}\|_2 \tag{59}$$

$$\leq \frac{\log m}{\sqrt{m}} \left( \sum_{\gamma=1}^{\beta} \left( \frac{\log m}{\sqrt{m}} \|\mathbf{N}\boldsymbol{v}\|_2 \right)^2 \right)^{1/2} \tag{60}$$

$$\leq 2 \frac{\log^2 m}{\sqrt{m}} \|\boldsymbol{v}\|_2. \tag{61}$$

Therefore, we have the estimate of $\mathrm{x}_\gamma^{(0)}$.

Next, we will estimate the 2nd term $\mathrm{x}_\gamma^{(1)}$. By definition, we may write

$$\sum_{\gamma=1}^{\beta} \mathrm{x}_\gamma^{(1)} = \sum_{\gamma=1}^{\beta} \widetilde{\zeta}_\gamma [\mathbf{dW}^l]_{i,j_\gamma} \mathrm{h}_\gamma. \tag{62}$$

Note that we use Theorem 1 (the convergence theorem) to write

$$\|[\mathbf{dW}^l]_i\|_2 = \|[\mathbf{W}^{l,(t)}]_i - [\mathbf{W}^l]_i\|_2 \leq O\left( \lambda \frac{\log m}{m} \right). \tag{63}$$

Thus, applying the estimate of $\mathrm{h}_\gamma$ to eq. (58), we have

$$\left| \sum_{\gamma=1}^{\beta} \mathrm{x}_\gamma^{(1)} \right| \leq 2 \sum_{\gamma=1}^{\beta} \left( \frac{\log m}{\sqrt{m}} \|\boldsymbol{v}\|_2 \right) [\mathbf{dW}^l]_{i,j_\gamma} \tag{64}$$

$$\leq \frac{\log m}{\sqrt{m}} \|\boldsymbol{v}\|_2 \cdot \sqrt{\beta} \; \|[\mathbf{dW}^l]_i\|_2 \tag{65}$$

$$\leq C\lambda \frac{\log^2 m}{m} \|\boldsymbol{v}\|_2 \leq \frac{\log^2 m}{\sqrt{m}} \|\boldsymbol{v}\|_2. \tag{66}$$

The last inequality uses $\lambda \leq o(\sqrt{m})$, which is indicated by the result of Theorem 1, therefore we have the estimate of $\mathrm{x}_\gamma^{(1)}$.

Next, we will estimate the 3rd term $\mathrm{x}_\gamma^{(2)}$. By definition, we may write

$$\sum_{\gamma=1}^{\beta} \mathrm{x}_\gamma^{(2)} = \sum_{\gamma=1}^{\beta} \widetilde{\zeta}_\gamma [\mathbf{W}^l]_{i,j_\gamma} \mathrm{dh}_\gamma. \tag{67}$$

We set $\mathrm{u}_\gamma^{(2)} := \widetilde{\zeta}_\gamma \mathrm{dh}_\gamma$ and $\mathbf{u}^{(2)} := (\mathrm{u}_1^{(2)}, \ldots, \mathrm{u}_\beta^{(2)})$. Using the randomness of $\mathbf{W}^l$ and general Hoeffding's inequality (Vershynin, 2018), for each fixed vector $\boldsymbol{v}$, we have

$$\mathbb{P} \left\{ \left| \sum_{\gamma=1}^{\beta} \mathrm{u}_\gamma^{(2)} [\mathbf{W}^l]_{i,j_\gamma} \right| > \tau \right\} \leq 2 \exp\left( -\frac{C\tau^2}{\|\mathbf{u}^{(2)}\|_2^2 \; \max_\gamma^2 \|[\mathbf{W}^l]_{i,j_\gamma}\|_{\psi_2}} \right) \tag{68}$$

$$\leq 2 \exp\left( -\frac{Cm\tau^2}{2\|\mathbf{u}^{(2)}\|_2^2} \right) \tag{69}$$

Note that

$$\mathrm{dh}_\gamma = \sum_{\rho=1}^{m} [\mathbf{dW}^{l-1}]_{j_\gamma,\rho} [\mathbf{N}\boldsymbol{v}]_\rho, \tag{70}$$

and choose $\tau := \frac{\log m}{\sqrt{m}}\|\mathbf{u}^{(2)}\|_2$, with probability $1 - e^{-\Omega(\log^2 m)}$, we have

$$\left|\sum_{\gamma=1}^{\beta} \mathrm{x}_{\gamma}^{(2)}\right| \leq \frac{\log m}{\sqrt{m}}\|\mathbf{u}^{(2)}\|_2 \leq \frac{\log m}{\sqrt{m}}\left\{\sum_{\gamma=1}^{\beta}\left(\sum_{\rho=1}^{m}[\mathbf{dW}^{l-1}]_{j_\gamma,\rho}[\mathbf{N}\boldsymbol{v}]_\rho\right)^2\right\}^{1/2} \tag{71}$$

$$\leq \frac{\log m}{\sqrt{m}}\left(\sum_{\gamma=1}^{\beta}\|[\mathbf{dW}^{l-1}]_{j_\gamma}\|_2^2\|\boldsymbol{v}\|_2^2\right)^{1/2} \leq \frac{\log m}{\sqrt{m}}\|\mathbf{dW}^{l-1}\|_F\|\boldsymbol{v}\|_2 \tag{72}$$

$$\leq \frac{\log^2 m}{\sqrt{m}}\|\boldsymbol{v}\|_2. \tag{73}$$

The last inequality uses $\lambda \leq o(\sqrt{m})$ and the result of Theorem 1:

$$\|\mathbf{dW}^{l-1}\|_F = \|\mathbf{W}^{l-1,(t)} - \mathbf{W}^{l-1}\|_F \leq O\left(\lambda\frac{\log m}{\sqrt{m}}\right). \tag{74}$$

Therefore, we have the estimate of $\mathrm{x}_\gamma^{(2)}$.

Next, we will estimate the last term $\mathrm{x}_\gamma^{(3)}$. Note that since $\widetilde{\zeta}_\gamma = \pm 1$, we have $|\widetilde{\zeta}_\gamma| = 1$, and conditioning on this event (56) happens, we may write

$$\left|\sum_{\gamma=1}^{\beta} \mathrm{x}_{\gamma}^{(3)}\right| = \left|\sum_{\gamma=1}^{\beta} \widetilde{\zeta}_\gamma[\mathbf{dW}^l]_{i,j_\gamma}\mathrm{dh}_\gamma\right| \tag{75}$$

$$\leq \|[\mathbf{dW}^l]_i\|_2\left\{\sum_{\gamma=1}^{\beta}\left(\sum_{\rho=1}^{m}[\mathbf{dW}^{l-1}]_{j_\gamma,\rho}[\mathbf{N}\boldsymbol{v}]_\rho\right)^2\right\}^{1/2} \tag{76}$$

$$\leq \|[\mathbf{dW}^l]_i\|_2\left(\sum_{\gamma=1}^{\beta}\|[\mathbf{dW}^{l-1}]_{j_\gamma}\|_2^2\|\boldsymbol{v}\|_2^2\right)^{1/2} \tag{77}$$

$$\leq \|[\mathbf{dW}^l]_i\|_2\|\mathbf{dW}^{l-1}\|_F\|\boldsymbol{v}\|_2 \leq \frac{\log^2 m}{\sqrt{m}}\|\boldsymbol{v}\|_2. \tag{78}$$

Putting this all together, with probability $1 - e^{-\Omega(\log^2 m)}$, we have

$$\left|\frac{d}{ds}\breve{g}_i^l(s) - \frac{d}{ds}\widetilde{g}_i^l(s)\right| = \left|\sum_{\gamma=1}^{\beta}\mathrm{x}_\gamma\right| \leq 4\frac{\log^2 m}{\sqrt{m}}\|\boldsymbol{v}\|_2. \tag{79}$$

Therefore, we have

$$|\breve{g}_i^l(s) - \widetilde{g}_i^l(s)| \leq \int_0^s\left|\frac{d}{ds}\breve{g}_i^l(s) - \frac{d}{ds}\widetilde{g}_i^l(s)\right|ds \tag{80}$$

$$\leq 4s\frac{\log^2 m}{\sqrt{m}}\|\boldsymbol{v}\|_2 \quad (0 \leq \forall s \leq 1). \tag{81}$$

This finishes the proof of (7) in Lemma 3. Finally, the proof of (8) in Lemma 3 is identical to the above proof, except the fact that each entry of $\mathbf{B}$ follows from $\mathcal{N}(0, 2/c)$ instead of $\mathcal{N}(0, 2/m)$. We complete the proof.

$\square$

## A.2 Proof of Generalization Theorem

**Theorem 2** (reshown, see §3)**.** *Under the same setting as Theorem 1, with probability at least* $1 - e^{-\Omega(\log^2 m)}$*, for every* $\boldsymbol{x}^{(p)}, \boldsymbol{x}^{(q)}$ ($p, q \in [n]$ $p \neq q$)*,* $t \in [T]$*,* $l \in [L]$*,* $i \in [m]$ *and* $j \in [c]$*, we*

*have*

$$\sup_{0 \le s \le 1} \left| g_i^{l,(t)}(\boldsymbol{x}(s)) - \tilde{g}_i^{l,(t)}(s) \right| \le O\left( \frac{\log^l m}{\sqrt{m}} \right) \|\boldsymbol{v}\|_2, \tag{82}$$

$$\sup_{0 \le s \le 1} \left| f_j^{(t)}(\boldsymbol{x}(s)) - \tilde{f}_j^{(t)}(s) \right| \le O\left( \frac{\log^{L+1} m}{\sqrt{c}} \right) \|\boldsymbol{v}\|_2. \tag{83}$$

In order to give provable guarantees for the generalization performance of over-parameterized deep networks in the $l^2$ regression task on the one-dimensional input-space, we apply the above Theorem 2 to bound the error between the trained network output $f^{(T)}(\boldsymbol{x})$ and the piecewise linear function $\hat{f}(\boldsymbol{x})$. Note that if we apply Theorem 1 to this case (one-dimensional models), simple preprocessing is needed: It just consider a mapping $[0, \nu] \to S_+^1$ (a half circle).

$$S_+^1 = \{ x = (x_1, x_2) \in \mathbb{R}^2 : \|x\|_2 = 1, x_2 \ge 1 \}$$

It is apparent that there exists a $C^\infty$ diffeomorphism from $[0, \nu]$ to $S_+^1$. Under this setting, the input from $S_+^1$ is the two-dimension. More precisely, the input needs another axis of coordinates to express the biases term. This procedure is written in the subsection of Dataset and loss function in the paper p.4. As a result, we deal with not the case of $d = 1$ but the case of $d = 3$ in Theorem 1.

**Lemma 5.** *Under the same setting as Theorem 1 and 2, for $\delta \in (0, 1/2]$, then with probability at least $1 - (\delta + e^{-\Omega(\log^2 m)})$, we have*

$$\sup_{x \in [0, \nu]} \left| f_j^{(T)}(x) - \hat{f}_j(x) \right| \le O\left( \frac{\log^{L+1} m}{m^{1/6}} \cdot \frac{\nu}{n\delta} \right). \tag{84}$$

*Proof.* Without loss of generality, we may assume that the entry of the weight matrix $B_{i,j} \sim \mathcal{N}(0, 2/\mathfrak{c})$, where $\mathfrak{c} = \Theta\left( m^{1/3} \right)$.[6]

For $k \in [n+1]$, we denote by $|\mathbb{I}_k|$ the length of the interval $\mathbb{I}_k = \left[ x^{(k-1)}, x^{(k)} \right]$. We set $c = \mathfrak{c} = m^{1/3}$ in Theorem 2, with probability at least $1 - e^{-\Omega(\log^2 m)}$, we have

$$\sup_{x \in \mathbb{I}_k} \left| f_j^{(T)}(x) - \tilde{f}_j^{(T)}(x) \right| \le C_j \frac{\log^{L+1} m}{m^{1/6}} \left( \sup_{k \in [n+1]} \left| x^{(k)} - x^{(k-1)} \right| \right) = C_j \frac{\log^{L+1} m}{m^{1/6}} \mathcal{T} \tag{85}$$

where $C_j$ is a constant, and $\mathcal{T} := \sup_{k \in [n+1]} \left| x^{(k)} - x^{(k-1)} \right|$. Here we set $\varepsilon := C_j \frac{\log^{L+1} m}{m^{1/6}} \mathcal{T}$.

Note that $f_j^{(T)}(x^{(k-1)}) = \hat{f}_j(x^{(k-1)})$ and $f_j^{(T)}(x^{(k)}) = \hat{f}_j(x^{(k)})$, by eq. (85) we have

$$\left| \tilde{f}_j^{(T)}(x^{(k-1)}) - \hat{f}_j(x^{(k-1)}) \right| \le \varepsilon \quad \text{and} \quad \left| \tilde{f}_j^{(T)}(x^{(k)}) - \hat{f}_j(x^{(k)}) \right| \le \varepsilon. \tag{86}$$

Note that the piecewise linear function $\hat{f}_j$ linearly connects $\left( x^{(k-1)}, \hat{f}_j(x^{(k-1)}) \right)$ and $\left( x^{(k)}, \hat{f}_j(x^{(k)}) \right)$. Thus, the following inequality holds.

$$\sup_{x \in \mathbb{I}_k} \left| \tilde{f}_j^{(T)}(x) - \hat{f}_j(x) \right| \le \varepsilon \tag{87}$$

We use eq. (85) and eq. (87) to see that

$$\sup_{x \in \mathbb{I}_k} \left| f_j^{(T)}(x) - \hat{f}_j(x) \right| \le \sup_{x \in \mathbb{I}_k} \left| f_j^{(T)}(x) - \tilde{f}_j^{(T)}(x) \right| + \sup_{x \in \mathbb{I}_k} \left| \tilde{f}_j^{(T)}(x) - \hat{f}_j(x) \right| \le 2\varepsilon. \tag{88}$$

Note that $\varepsilon$ is not dependent on $k$. Thus, we have

$$\sup_{x \in [0, \nu]} \left| f_j^{(T)}(x) - \hat{f}_j(x) \right| \le 2\varepsilon. \tag{89}$$

---

[6] For each output $y = (y_1, y_2, \cdots, y_c)$, an additional coordinates $(y_{c+1}, y_{c+2}, \ldots, y_{\mathfrak{c}}) = (0, 0, \ldots, 0)$ can always be padded to the output. We did not try hard to improve the exponent $1/3$.

Next, we evaluate $\mathcal{T} = \sup_{k \in [n+1]} \left| x^{(k)} - x^{(k-1)} \right|$. Recall that the training dataset $\{x^{(k)}\}_{k \in [n]}$ is generated from the uniform distribution in the interval $[0, \nu]$. We denote by $\mu := n/\nu$ the density of training samples. In this setting, it is well known that for every $r > 0$ ($r < \nu$), the number of training samples in the interval $[0, r]$ follows the Poisson distribution with mean $\mu r$, then the sequence of inter-sample distances is independent and identically distributed exponential random variables having mean $1/\mu$ ($= \frac{\nu}{n}$) (Daley & Vere-Jones, 2007). Define $\Delta$ to be the above exponential distribution having mean $\nu/n$, which is the inter-sample distance. Then, using Markov's inequality, for $\tau > 0$, we have

$$\mathbb{P}[\Delta \geq \tau] \leq \frac{\mathbb{E}[\Delta]}{\tau} = \frac{\nu}{n\tau}. \tag{90}$$

Thus, we can choose $\tau := \frac{\nu}{n\tau}$, which implies that with probability at least $1 - \delta$, we have

$$\mathcal{T} \leq \frac{\nu}{n\delta}. \tag{91}$$

Combining this with eq. (89), with probability at least $1 - (\delta + e^{-\Omega(\log^2 m)})$, we have

$$\sup_{x \in [0, \nu]} \left| f_j^{(T)}(x) - \hat{f}_j(x) \right| \leq 2C_j \frac{\log^{L+1} m}{m^{1/6}} \mathcal{T} \leq 2C_j \frac{\log^{L+1} m}{m^{1/6}} \frac{\nu}{n\delta}. \tag{92}$$

We complete the proof of the Lemma 5. $\qquad\square$

**Lemma 6.** *Suppose $f_j^*(x)$ is a $C^1$- class function on $[0, \nu]$ and $\hat{f}_j(x)$ is the linear interpolation of the data points $\left\{ \left( x^{(i)}, f_j^*(x^{(i)}) \right) \right\}_{i \in [n]}$. For $\delta \in (0, 1/2]$, with probability at least $1 - \delta$, we have*

$$\sup_{x \in [0, \nu]} \left| \hat{f}_j(x) - f_j^*(x) \right| \leq O\left( \frac{\nu}{n\delta} \right). \tag{93}$$

*Proof.* We use $f_j^{*\prime}(x)$ to denote the first derivative of $f_j^*(x)$. Since $f_j^*$ is a $C^1$- class function on the compact set $[0, \nu]$, $f_j^{*\prime}$ is a bounded function. Thus, we can define $\mathcal{S}^{(k)}$ and $\mathcal{S}$ as follows:

$$\mathcal{S}^{(k)} := \sup_{x \in \mathbb{I}_k} \left| f_j^{*\prime}(x) \right|, \quad \mathcal{S} := \sup_{k \in [n+1]} \mathcal{S}^{(k)}. \tag{94}$$

Since $f_j^*(x)$ and $\hat{f}_j(x)$ are continuous functions, there exists $\mathfrak{h}^{(k)}$ ($0 \leq \mathfrak{h}^{(k)} \leq x^{(k)} - x^{(k-1)}$) such that

$$\sup_{x \in I_k} \left| f_j^*(x) - \hat{f}_j(x) \right| = \left| f_j^*(x^{(k-1)} + \mathfrak{h}_k) - \hat{f}_j(x^{(k-1)} + \mathfrak{h}_k) \right|. \tag{95}$$

By the mean value theorem, there exists a real number $\mathfrak{c}^{(k)} \in \mathbb{I}_k$ such that,

$$\hat{f}_j(x^{(k-1)} + \mathfrak{h}^{(k)}) = f_j^{*\prime}(\mathfrak{c}^{(k)}) \mathfrak{h}^{(k)} + f_j^*(x^{(k-1)}). \tag{96}$$

Note that by the definition of $\hat{f}_j$, we have $f_j^*(x^{(k-1)}) = \hat{f}_j(x^{(k-1)})$, therefore, we also have that

$$\left| f_j^*(x^{(k-1)} + \mathfrak{h}^{(k)}) - \hat{f}_j(x^{(k-1)} + \mathfrak{h}^{(k)}) \right| = \left| f_j^*(x^{(k-1)} + \mathfrak{h}^{(k)}) - f_j^*(x^{(k-1)}) - f_j^{*\prime}(\mathfrak{c}^{(k)}) \mathfrak{h}^{(k)} \right| \tag{97}$$

$$\leq \int_{x^{(k-1)}}^{x^{(k-1)} + \mathfrak{h}^{(k)}} \left| f_j^{*\prime}(x) - f_j^{*\prime}(\mathfrak{c}^{(k)}) \right| dx \tag{98}$$

$$\leq 2\mathfrak{h}^{(k)} \mathcal{S}^{(k)}. \tag{99}$$

By considering whole interval $[0, \nu]$, we have

$$\sup_{x \in [0, \nu]} \left| \hat{f}_j(x) - f_j^*(x) \right| = \sup_{k \in [n+1]} \left( \sup_{x \in I_k} \left| f_j^*(x) - \hat{f}_j(x) \right| \right) \leq 2\mathcal{T}\mathcal{S} \tag{100}$$

Now we apply the estimate of $\mathcal{T}$ from eq. (91). We may therefore rewrite eq. (100) as

$$\sup_{x \in [0, \nu]} \left| \hat{f}_j(x) - f_j^*(x) \right| \leq O\left( \frac{\nu}{n\delta} \right). \tag{101}$$

We complete the proof of Lemma 6.

$\qquad\square$

**Proof of Theorem 4 (Generalization)**

*Proof.* From eq. (89) and eq. (93), with probability at least $1 - (\delta + e^{-\Omega(\log^2 m)})$, we have for every $j = 1, 2, \cdots, c$,

$$
\left| f_j^{(T)}(x) - f_j^*(x) \right| \leq \left| f_j^{(T)}(x) - \hat{f}_j(x) \right| + \left| \hat{f}_j(x) - f_j^*(x) \right|
$$

$$
\leq C_j \frac{\nu}{n\delta} \left( 1 + \frac{\log^{L+1} m}{m^{1/6}} \right) \qquad (\forall x \in [0, \nu]).
$$

Using the triangle inequality, this gives

$$
\ell \left( f_j^{(T)}(x), y \right) = \frac{1}{2} \left\| f^{(T)}(x) - f^*(x) \right\|_2^2 = \frac{1}{2} \sum_{j=1}^{c} \left\| f_j^{(T)}(x) - f_j^*(x) \right\|^2
$$

$$
\leq \frac{1}{2} C'^2 c \frac{\nu^2}{n^2 \delta^2} \left( 1 + \frac{\log^{L+1} m}{m^{1/6}} \right)^2 \qquad (\forall x \in [0, \nu]).
$$

where $C' := \max_{j=1,\cdots,c} C_j$.

Thus, we have the following estimate:

With probability at least $1 - (\delta + e^{-\Omega(\log^2 m)})$, we have

$$
\mathbb{E}_{(x,y) \sim \mathcal{D}} \left[ \ell \left( f^{(T)}(x), y \right) \right] = \frac{1}{\nu} \int_0^\nu \frac{1}{2} \left\| f^{(T)}(x) - f^*(x) \right\|^2 dx \tag{102}
$$

$$
\leq \frac{1}{2} C'^2 c \frac{\nu^2}{n^2 \delta^2} \left( 1 + \frac{\log^{L+1} m}{m^{1/6}} \right)^2. \tag{103}
$$

We complete the proof. □

