# OpenReview forum: "A Mechanism of Implicit Regularization in Deep Learning"
_ICLR.cc/2020/Conference — Reject_

### Official Review · AnonReviewer1 · 2019-10-22
**Official Blind Review #1**

**Rating:** 3

**Review:**

This paper suggests a new technique to analyze the implicit regularization caused by ReLU activations. They bound the generalization error by two terms: 1) one term that represents the distance between the trained network output and a piecewise linear function built based on the set of training points and 2) another term that represents the distance between the piecewise linear approximation and the desired target. The first term is bounded using a random walk type of analysis, which to the best of my knowledge is novel.
I find this technique rather interesting and technically sound, although I do have a number of concerns and I'm at the moment more on the reject side, although I will re-consider my score if the authors can provide satisfactory answers.

Generalization to more complex activation functions
If I understand correctly, the interpolation technique between two points only works for ReLU functions. If one were to try to generalize the analysis to more complex non-linear functions by using a more complex interpolation schemes, wouldn’t you then have a random walk in high-dimensions? If so, wouldn’t that be a problem given the different properties of Brownian motion in high-dimensions?

Generalization to smooth activation functions
Another question related to the previous one is whether one could hope to generalize the analysis to smooth activation functions. I believe this is also a drawback of combinatorial techniques such as Hanin and Rolnick which have to rely on the discrete nature of the breakpoints.

Generalization bound is only derived for 1-d functions
Theorem 2 is derived for each dimension independently while the generalization results in Theorem 4 are for 1-dimensional inputs. Where is the difficulty in generalizing these results to higher dimensions?

Prior work on generalization of SGD
I was really expecting a discussion about how the generalization bound derived in this paper compares to prior work, e.g.
Hardt, Moritz, Benjamin Recht, and Yoram Singer. "Train faster, generalize better: Stability of stochastic gradient descent." arXiv preprint arXiv:1509.01240 (2015).
Kuzborskij, Ilja, and Christoph H. Lampert. "Data-dependent stability of stochastic gradient descent." arXiv preprint arXiv:1703.01678 (2017).
Brutzkus, Alon, et al. "Sgd learns over-parameterized networks that provably generalize on linearly separable data." arXiv preprint arXiv:1710.10174 (2017).
And many others…
For instance the bound derived in Hardt et al. is also of the order O(n^-2). The bounds in Kuzborskij are also data-dependent and so are yours since your generalization bound depends on the density of the training points. Can you comment on this? What specific insights do we gain your analysis?

Noise SGD
My understanding is that the authors assume that the noise of SGD is Gaussian. Although this is commonly used when analyzing SGD, there is evidence that the noise is actually not Gaussian, see e.g.
Daneshmand, Hadi, et al. "Escaping saddles with stochastic gradients." arXiv preprint arXiv:1803.05999 (2018).
Simsekli, Umut, Levent Sagun, and Mert Gurbuzbalaban. "A tail-index analysis of stochastic gradient noise in deep neural networks." arXiv preprint arXiv:1901.06053 (2019).
I feel this is worth pointing out and one could perhaps also extend this analysis to Heavy-tail noise. I would expect that the results would still hold in expectation but perhaps with a slightly worse probability.

Influence step size SGD
Using larger step sizes in the SGD updates increase the influence of the noise. I was expecting this to somehow be captured in your analysis but I fail to see where it appears. Can you comment on this?

Proof Lemma 3
The derivation of Eq. 10 does not seem completely justified in the proof in the appendix. The authors essentially prove that the length of the gradient gap is bounded by |S| but why is the coefficient \omega distributed according to a normal distribution. It seems to me that you need the noise of SGD to be Gaussian for such statement to hold. Can you confirm this? If so, I think this needs to be clearly stated as an assumption since -- as pointed out above -- this is not necessarily true in practice.

Minor: proof Theorem 2
It seems rather trivial but for completeness, you should write the proof of Eq. (6) in Theorem 2.

“A priori estimates”
This is a terminology that is often used in the paper but never defined. What do you mean by “a priori” in this context?

Minor comment
I would move footnote 3 directly in the main step. I think it is important to point out that the steps of the random walk correspond to the breakpoints.


**Experience Assessment:**

I have published one or two papers in this area.

**Review Assessment: Checking Correctness Of Derivations And Theory:**

I carefully checked the derivations and theory.

**Review Assessment: Checking Correctness Of Experiments:**

N/A

**Review Assessment: Thoroughness In Paper Reading:**

I read the paper thoroughly.

---

> ### Author Response · Authors · 2019-11-13
> **Response to Reviewer #1 (part1)**
>
> Thank you so much for your very constructive comments! Please see below for our responses to your comments.
>
> >> Generalization to more complex activation functions  and
> >> Generalization to smooth activation functions
>
> We applied the technique only to ReLU networks, but we think that it is possible to expand this technique to the case of Leaky ReLU function.
> Our technique is not applicable and does not work in the smooth activation functions.
>
> We introduced gradient gaps of neural networks, which is equivalent to calculating the second order derivative of neural network functions. As for smooth activation functions, it is thus better to directly calculate the second order derivatives of neural networks.
>
> As you pointed out, it is difficult to analyze neural networks for non-smooth activation functions other than ReLU and Leaky ReLU.
>
>
> >>> Generalization bound is only derived for 1-d functions
>
> The main purpose of our work is to understand the mechanism of implicit regularization and to clarify how it controls considerably large capacity of deep neural networks.
>
> As a result of understanding a mechanism of implicit regularization, we obtained a generalization bound for over-parameterized DNNs (Theorem 4), which we will generalize to high-dimension regimes.
>
> In Theorem 2, we prove low complexity of DNN in high-dimension regimes, which is derived from a mechanism of implicit regularization in DNNs.
>
> When SGD finds the global minimum on the training objective, DNNs fit all training points. In order to obtain generalization bounds as SGD have converged, we have to analyze the output of DNNs between each training point.
>
> Eq. (5) shows that once the DNN is trained and converged by SGD with a sufficiently large step size, the output of each hidden neuron in over-parameterized DNNs is very close to the piecewise linear interpolation of the training points.
> From this result, it turns out that DNNs have low complexity in that each hidden neuron have to connect sample points almost straight.
>
> In Allen-Zhu et al., 2018a it proved that the Rademacher complexity of two and three-layer ReLU neural networks is small using a different technique from us.
>
> Generalizing to high-dimension regimes could be achieved by applying formula (5) to prove that the empirical Rademacher complexity of DNNs is small.
>
> In this work, we clarified a mechanism of implicit regularization in DNNs by proving that the output of DNNs is very close to the piecewise linear interpolation of the training points and by obtaining generalization bounds based on this low complexity of DNNs.
> In order to explain clearly that DNN approximately convergent to piecewise linear interpolation between data points, we chose the one-dimensional regression problems.
>
> We note that generalization bounds for high-dimension regimes are based on the fact that the output of DNNs is very close not to the piecewise linear interpolation but to the test points for the empirical Rademacher complexity.
> It is important to generalize to high-dimension regimes, but we leave this as a suggested direction for future research.

---

> ### Author Response · Authors · 2019-11-13
> **Response to Reviewer #1 (part2)**
>
> We appreciate your detailed reading of the paper and thoughtful comments. We have responded to these below.
>
> >>> Prior work on generalization of SGD
>
> In Both Hardt et al. (arXiv:1509.01240) and In Kuzborskij et al. (arXiv:1703.01678), it proved generalization bounds of order \(O(n^{-1/2})\) for not over-parameterized NNs.
>
> In particular, generalization bounds of NNs with low Rademacher complexity is the representative result of statistical learning theory.
>
> Since the number of neurons in over-parameterized DNNs is much larger than the number of training points, the classical technique of statistical learning theory is not applicable.
>
> Brutzkus et al. (arXiv:1710.10174) and Li & Liang (arXiv:1808.01204) proved convergence rates of SGD to a global minima and obtain generalization bound of order \(O(n^{-1/2})\) studying the problem of learning a two-layer over-parameterized neural networks with linearly separable or structured data.
>
> In this paper, we study over-parameterized deep neural network and obtain the generalization bounds of order \(O(n^{-2})\) for one-dimensional inputs. For high-dimension regimes, the order of generalization bounds is \(O(n^{-1/2})\), which is due to applying Rademacher complexity.
>
> DNNs exhibit good generalization performance in over-parameterized regime, where the number of neurons in the network exceeds the number of training points, and the cause of the phenomenon is thought to be that some mechanisms of implicit regularization properly control rich expressibility of over-parameterized DNNs.
>
> Nonetheless, a direct and concrete mechanism of implicit regularization has not been clarified.
>
> Recent works use the Rademacher complexity for proving generalization bounds of over-parameterized neural networks (Allen-Zhu et al. 2018a and Arora et al. 2019b), and usually do not explain how the SGD favors low complexity solutions.
>
> Until recently, implicit regularization was thought to be expressed by minimizing some norm, which is similar to weight decay to minimize (\L^2\) norm.
>
> For example, in Gunasekar et al. (arXiv:1705.09280), the authors conjectured the implicit regularizer as the nuclear norm and proved with some assumption that gradient descent converges to the minimum nuclear norm solution.
>
> However, this way of understanding implicit regularization was denied by Arora et al. 2019a.
>
> One of our main results is to provide a new interpretation of implicit regularization for over-parameterized neural networks.
>
>
> >>> Noise SGD
>
> In this paper, we do not assume that the noise of SGD is Gaussian. Allen-Zhu et al. 2018b proved upper bounds on the amount of the weight change by SGD, and we used the bounds to prove Theorem 2 and Theorem 4.
> The bounds are in Theorem 1 (Allen-Zhu et al. 2018b) as follows:
> \[ \| W^{t} - W^{0} \| \leq O\left( \lambda \frac{\log m}{ \sqrt{m}} \right) \]
>
> In the over-parameterized regime, as shown in Theorem 1, the weight change before and after training is small enough, so we can analyze SGD without assuming that noise of SGD is Gaussian.
>
> We certainly think that Theorem 2 and Theorem 4 could be made better by studying SGD more carefully, but that is an issue for future work.
>
>
> >>> Influence step size SGD
>
> In Theorem 1 (Allen-Zhu et al. 2018b), \(\eta\) is the step size (learning rate). As you indicate, using larger \(\eta\) increases the weight change, which results in worse generalization bounds.

---

> ### Author Response · Authors · 2019-11-13
> **Response to Reviewer #1 (part3)**
>
> We appreciate your detailed reading of the paper and thoughtful comments. We have responded to these below.
>
> >>> Proof Lemma 3
>
> Let us explain that the coefficient \(\omega\) is close to the initial value of the corresponding weight and \(\omega\) is not a noise of SGD.
>
> The following is the discussion about \(l\)-th layer, and we drop the subscript \(l\) for notational convenience.
>
> \(x_{\gamma}\) is expressed as Eq. (44) in p.12.
> \[ x_{\gamma} = \left[ W^{(t)} \right]_{i,j} \left[ M^{(t)}v \right]_{j} \xi ~~ \tag{44} \]
>
> We use \(W^{t}\) to denote a weight matrix for \(l\)-th layer after \(t\) iterations of SGD. In Eq. (46), we define the difference between \(W^{t}\) and initial value \(W^{0}\) as follows:
> \[ dW :=  W^{t} - W^{0} \tag{46} \]
>
> This is the weight change by SGD, and Theorem 1 guarantees that \(\| dW \|\) is small enough. That is, \(\| W^{t} - W^{0} \| = \| dW \| \leq \varepsilon\), where \(\varepsilon >0\) is enough small related to the network width.
>
> For notational convenience, let \(S\) be \(\left| \left[ M^{(t)}v \right]_{j} \xi \right|\). We obtain the following upper bounds from Eq. (58) and Eq. (56).
> \[ \left| \left[ M^{(t)}v \right]_{j} \xi \right| = \left| S \right| \leq \frac{\log m}{\sqrt{m}}\| v \|\]
>
> From the above, Eq. (44) can be expressed as follows:
> \[ x_{\gamma} = S\left(\left[ W^{(0)} \right]_{i,j} + \varepsilon \right) \]
>
> Each weight value is initialized by He Normal, so \(\left[ W^{(0)} \right]_{i,j} \sim \mathcal{N}(0, \frac{2}{m})\). Thus, because \(\left[ W^{(0)} \right]_{i,j}\) is Gaussian random value and \(\varepsilon >0\) is small enough, the sum of \(x_{\gamma}\) is `Gaussian random walk'.
>
> From the above discussion, you can see that the coefficient \(\omega\) is not a noise of SGD but the initial value of a weight.
>
>
> >>> proof Theorem 2
>
> Thank you for your advice. The output dimension of the last layer is \(c\) while that of other layers is \(m\), so we prove Eq. (6) by changing inequality in the proof of Eq. (5). We already changed the paper, so please confirm.
>
>
> >>> “A priori estimates”
>
> Thank you for your advice. We have modified our paper to clarify what we wanted to convey by using the terminology "a priori estimates".
>
> An a priori estimate is used in the theory of partial differential equations, but already used in machine theory for example:
> Weinan E et al., "A Priori Estimates for Two-layer Neural Networks", (arXiv:1810.06397).
>
> A priori is Latin for "from before" and refers to the fact that the estimate for the solution of minimizing an objective function is derived before the solution is known to exist.
>
> In other words, we show that in Theorem 2, the difference between the network output function and its linear interpolation is evaluated only from the amount of the weight change.
>
>
> >>>  footnote 3
>
> Thank you for your advice. We moved footnote 3 in the main step. Please confirm.

---

### Official Review · AnonReviewer2 · 2019-10-23
**Official Blind Review #2**

**Rating:** 1

**Review:**

This paper studies the implicit regularization in deep learning under the over-parameterized setting. In specific, the authors study the neural network outputs and “pre-activation values” on line segments connecting two training data inputs and characterize an implicit regularization based on it. I have the following concerns:

First of all, it is not clear to me why Theorem 2 is relevant to “implicit regularization”. To my knowledge, implicit regularization or implicit bias statements in prior works cited in this paper are all about the convergence to a specific solution for underdetermined problems. For example, “among all solutions that fits the data, gradient descent converges to minimum distance solution to initialization (linear model square loss), maximum margin solution (linear model exponential loss), minimum nuclear norm solution (matrix sensing with small initialization)”. In comparison, Theorem 2 just gives some bounds that holds for every sgd iteration. I cannot see any connection between Theorem 2 and implicit regularization.

Moreover, the authors’ claim “the implicit regularization in over-parameterized DNNs has not been identified” is not correct. As the authors mentioned,  the neural network is close to its linear approximation model with respect to weight parameters at initialization. Therefore the implicit bias of (stochastic) gradient descent for DNNs in the over-parameterized regime is essentially implicit bias of (stochastic) gradient descent for linear models (for square loss). In Arora et al., 2019b it has been proved that infinitely wide neural networks trained with gradient flow converges to the NTK-based kernel regression solution.  So at least for gradient flow with square loss, the implicit bias of DNNs has been well-studied. In fact in a missed reference [2], essentially the implicit bias for both gradient descent and stochastic gradient descent has been studied. The remark “in most cases, the authors used GD to derive their results by the NTK analysis” is also not convincing. Allen-Zhu et al., 2018a,b, Allen-Zhu & Li, 2019 and missed references [1,2,3,4] all studied SGD of over-parameterized neural networks, and some are not exactly in the so-called NTK regime. The authors should also compare their generalization bounds with existing results for SGD (Allen-Zhu et al., 2018a, Allen-Zhu & Li, 2019) and [3].

Finally, Theorem 4 only considers one-dimensional models, which is not a very interesting problem setting. Its proof might also be flawed. In fact, the setting in Section 4 is not consistent with Theorem 1, since Theorem 1 requires that all inputs have unit norm and their last coordinate should be a constant. For one dimensional case, this means all inputs must be the same scalar! Even if we ignore the last coordinate assumption in Theorem 1, for 1D case all inputs are still reduced to +1 or -1’s.


[1] Difan Zou, Yuan Cao, Dongruo Zhou, Quanquan Gu, Stochastic Gradient Descent Optimizes Over-parameterized Deep ReLU Networks
[2] Samet Oymak, Mahdi Soltanolkotabi, Overparameterized Nonlinear Learning: Gradient Descent Takes the Shortest Path?
[3] Yuan Cao, Quanquan Gu, Generalization Bounds of Stochastic Gradient Descent for Wide and Deep Neural Networks
[4] Difan Zou, Quanquan Gu, An Improved Analysis of Training Over-parameterized Deep Neural Networks






**Experience Assessment:**

I have published one or two papers in this area.

**Review Assessment: Checking Correctness Of Derivations And Theory:**

I assessed the sensibility of the derivations and theory.

**Review Assessment: Checking Correctness Of Experiments:**

N/A

**Review Assessment: Thoroughness In Paper Reading:**

I read the paper at least twice and used my best judgement in assessing the paper.

---

> ### Author Response · Authors · 2019-11-13
> **Response to Reviewer #2 (part1)**
>
> We appreciate your detailed reading of the paper and thoughtful comments. We have responded to these below.
>
> >> First of all, it is not clear to me why Theorem 2 is relevant to “implicit regularization”...
>
>
> Of course, implicit regularization could be shown as the convergence to minimum distance solution, maximum margin solution, or minimum nuclear solution.
>
> However, in Arora et al., 2019a it was shown that the weight dynamics of DNN by SGD cannot be represented as a kind of norm minimizing solutions. Our setting is also underdetermined problem, in other words, even if SGD learn a network to fit the training data, the network does not necessarily output properly for unknown inputs.
>
> Now, if it is proved that neural network outputs between data points are approximated by simple functions, what is the meaning of implicit regularization?
> Theorem 2 indicates that all hidden units (including outputs) approximately linearly interpolate between data points.
> As a result, the output functions of the learned networks by SGD is restricted to "simple" functions.
>
> In this meaning, it is clear that Theorem 2 is relevant to implicit regularization.
>
> We also make comments about the connection between Theorem 2 and implicit regularization in the response to reviewer #3, so please check it.
>
>
> >> Moreover, the authors’ claim “the implicit regularization in over-parameterized DNNs has not been identified” is not correct...
>
> Our claim is different from an assertion that nothing is known about implicit regularization in over-parameterized DNN.
> As written in Question 1, p.1 in this paper, we care about how the low complexity is caused by implicit regularization.
> Whether the DNN approximated by NTK has implicit bias is one thing and what kind of implicit regularization the DNN has is another.
>
> Of course, in Arora et al., 2019b it indicated that wide neural networks trained with gradient flow converges to the NTK-based kernel regression solution, and they proved generalization estimates.
>
> However, "what is" implicit regularization of DNN has not been clarified.
>
> Moreover, in the newer research Arora et al., 2019a showed that implicit regularization cannot be represented as a mathematical norm minimization problem, thus at least in the novel research, a mechanism of implicit regularization is an open problem and now many researchers study implicit regularization and generalization problem.
>
> Although whether implicit regularization is understood may depend on the researcher's stances, our studies are the first to mathematically characterize the connection between low complexity and implicit regularization.

---

> > ### Comment · AnonReviewer2 · 2019-11-15
> > **Re: Part 1**
> >
> > Thanks for your detailed response. I’m still not convinced by it on several issues I pointed out in my review.
> >
> > ---Although you provided some explanations, they did not directly clarify why Theorem 2 is not simply some sort of bounds, but an implicit bias result. If Theorem 2 is an implicit bias result, can we say that the bound on $\| \mathbf{w}_r(t) -  \mathbf{w}_r(0) \|_2$ in Lemma 3.3 in Du et al., 2018 is also an implicit bias result? Note that $\| \mathbf{w}_r(t) -  \mathbf{w}_r(0) \|_2$ also characterizes certain type of low complexity for the neural networks, and therefore matches your description on implicit bias.
> >
> > ---I am not convinced that the result in Arora et al., 2019a can demonstrate that implicit regularization is not understood in the NTK regime. It seems to me that the matrix factorization problem uses a scaling different from NTK. At least for square loss, I think the implicit bias of gradient descent in the NTK regime has been understood fairly well: if you know exactly the classifier your training algorithm gives you (Arora et al., 2019b), can you still say the implicit bias of your training algorithm is not understood?

---

> ### Author Response · Authors · 2019-11-13
> **Response to Reviewer #2 (part2)**
>
> We appreciate your detailed reading of the paper and thoughtful comments. We have responded to these below.
>
> >> In fact in a missed reference [2], essentially the implicit bias for both gradient descent and stochastic gradient descent has been studied. The remark “in most cases, the authors used GD to derive their results by the NTK analysis” is also not convincing...
>
>
> Unfortunately, none of the papers you suggest show the relation between generalization and implicit regularization in over-parameterized DNN. Especially, [2] focused on SGD convergence in various optimization problems.
>
> They applied their results to one hidden layer neural networks and did not show generalization bounds as they said; "our results do not directly address generalization" ( in the Supplementary PDF p.14 line 30, http://proceedings.mlr.press/v97/oymak19a/oymak19a-supp.pdf ).
>
> By contrast, our result shows that generalization bounds of DNN learned by SGD are proved by the piecewise linear interpolation between training data points. We consider this idea itself is novel enough to submit the paper.
>
> In order to explain clearly that DNN approximately convergent to piecewise linear interpolation between data points, we chose the one-dimensional regression problems. Generalizing to high-dimension regimes could be achieved by applying formula (5) to prove that the empirical Rademacher complexity of DNNs is small.
>
> Besides, none of the work has estimated generalization bounds by NTK for learning with STOCHASTIC GRADIENT DESCENT in the over-parameterized setting.
> In the infinite width limit, the linear model of NTK does not hold about the information of the network structure. In other words, for the NTK generalization bounds, we cannot investigate the effects of the number of neurons or the number of parameters.
>
>
> >> Finally, Theorem 4 only considers one-dimensional models, which is not a very interesting problem setting...
>
>
> As you point out, this part is certainly a little confusing, so we add some explanation to the Appendix.
>
> Certainly, when we apply Theorem 1 to one-dimensional cases, simple preprocessing will be needed: It just consider a mapping \([0,\nu] \to S^1_+\) (a half circle).
> \[ S^1_+ = \{ x=(x_1, x_2)\in \mathbb{R}^2 : \|x\|_2=1, x_2\geq 1 \} \]
> It is apparent that there exists a \(C^{\infty}\) diffeomorphism from \([0,\nu]\) to \(S^1_+\). Under this setting, the input from \(S^1_+\) is the two-dimension space. More precisely, the input needs another axis of coordinates to express the biases term. This procedure is written in the subsection of Dataset and loss function in the paper p.4.
>
> As a result, we deal with not the case of \(d=1\) but the case of \(d=3\) in Theorem 1.
>
> In this paper, we show that the neural network is approximated by a piecewise linear interpolation between data points via SGD training and clarify the connection between the interpolation and generalization bounds.
>
> To make clear the main points of the generalization bounds, we chose the one-dimensional regression task.
> It is important to generalize to high-dimension regimes, but we leave this as a suggested direction for future research.

---

> > ### Comment · AnonReviewer2 · 2019-11-15
> > **Re: Part 2**
> >
> > Thanks  for your detailed response. I’m still not convinced by it on several issues I pointed out in my review.
> >
> > ---Indeed [2] did not study generalization. But implicit bias and generalization are different things. In fact to my knowledge, Arora et al., 2019a and most other papers on implicit bias do not directly study generalization. My point is that it is not appropriate to claim that implicit bias of sgd in the NTK regime has not been studied. It is also inappropriate to claim “Although NTK analysis is limited to gradient descent, our analysis can be applied to stochastic gradient descent”.
> >
> > ---I do not agree with your comment “Besides, none of the work has estimated generalization bounds by NTK for learning with STOCHASTIC GRADIENT DESCENT in the over-parameterized setting.” The paper I pointed out in my review, “Generalization Bounds of Stochastic Gradient Descent for Wide and Deep Neural Networks”, studied exactly this! I think the proof of Lemma 4.2 in the paper (https://arxiv.org/pdf/1905.13210v2.pdf) is correct and you have misunderstood their proof.
> >
> > Equation (1) in your response is:
> > $$
> > L_i(W’) - L_i(W) = \ell[ y_i f_{W’}(x_i) ]  -  \ell[ y_i f_{W}(x_i) ]  \geq \ell’[ y_i f_{W}(x_i) ] \cdot y_i \cdot [ f_{W’}(x_i) - f_{W}(x_i) ]  \geq f_{W’}(x_i) - f_{W}(x_i).
> > $$
> > However, the last inequality above does not appear in the paper. The last two equations on page 15 in the paper (https://arxiv.org/pdf/1905.13210v2.pdf) are:
> > $$
> > L_i(W’) - L_i(W) = \ell[ y_i f_{W’}(x_i) ]  -  \ell[ y_i f_{W}(x_i) ]  \geq \ell’[ y_i f_{W}(x_i) ] \cdot y_i \cdot [ f_{W’}(x_i) - f_{W}(x_i) ]  ~~~~~~(*)
> > $$
> > and
> > $$
> > L_i(W’) - L_i(W) \geq \sum_{l=1}^L < \nabla_{W_l} L_i(W), W’_l - W_l   > - \ldots . ~~~~~~(**)
> > $$
> > Both of the above equations are correct, and to obtain (**) from (*), one can just apply Lemma 4.1 in the paper, and use the fact that $\ell’(\cdot)$ and $y_i$ are bounded, as is explained in the paper. This is because Lemma 4.1 essentially says that one can replace $ f_{W’}(x_i) - f_{W}(x_i) $ with $ < \nabla_{W} f_W(x_i) , W’ - W   >$ while only introducing a very small error. Bounds on $\ell’(\cdot)$ and $y_i$ also ensures that such error won’t be amplified. Therefore doing this replacement on the right-hand-side of (*) gives exactly (**).
> >
> > I think you have misunderstood the proof given in the paper and discovered “fatal flaws” based on your misunderstanding.

---

> > > ### Author Response · Authors · 2019-11-15
> > > **Re:Re: Response to Reviewer #2**
> > >
> > > Thank you for your answer and your interest, we value the discussion.
> > >
> > > We kindly disagree. Our claim is different from an assertion that implicit bias of sgd in the NTK regime has not been studied. In our paper, the NTK analysis indicates the equivalence theorem
> > > between the fully-trained wide neural network and the kernel regression predictor using the NTK, which is established by Arora et al. 2019b. We think that it is difficult to generalize to the SGD version of the equivalence theorem. However, as you point out, this part was certainly a little confusing, so we changed the paper, so please confirm.
> > >
> > > In contrast, we agree that the proof of the paper (https://arxiv.org/pdf/1905.13210v2.pdf) is correct, and we also changed the paper, so please confirm.
> > >
> > > Again, we thank the reviewer for the detailed comments and helpful suggestions. We hope our clarifications and explanations will convey a deeper understanding of our main ideas and major contributions.

---

### Official Review · AnonReviewer3 · 2019-10-23
**Official Blind Review #3**

**Rating:** 3

**Review:**

Review of "A mechanism of ... deep learning"

This paper studies the generalization performance and implicit regularization of deep learning. In particular, the authors propose a novel technique called "random walk analysis" to study the nonlinearity of the neural network with respect to the input data points. Moreover, the authors prove that for a class of 1-d continuously differentiable functions, SGD can achieve O(n^{-2}) generalization error bound.

Overall this paper is well written and easy to follow. The linear approximation with respect to input parameter space is also interesting and seems to be useful in the generalization analysis. Besides, I have the following comments and concerns.

1. I am a little bit confused by the definitions "Priori generalization estimates" and "posterior data distribution". I would like to see clearer description in the introduction.
2. I would like to see more discussion on Theorem 2 in the surrounding text, it is quite unclear to me why Theorem 2 is important and how it can be related to the "implicit regularization".
3. I do not see the proof of (6) in Section 3.2.
4. It seems that the generalization results in this paper are difficult to be generalized to high-dimension regimes. For example, if you assume that each entry of the training data point is generated from the uniform distribution in the interval [0, v], the density of training sample would be \mu = n/(v^d), and the resulting generalization bound would be O(v^d/(n\delta)), which is extremely large.
5. It seems that the generalization results hold for any data distribution. However, it is widely known that if the training data is randomly labeled, the neural network trained by SGD cannot achieve small population risk, which contradicts the result in Theorem 4.


----------------------------
After reading the authors' response, I still think that the generalization results in this paper are not significant and important, and how the theoretical results can be related to the implicit regularization. Thus I would like to keep my score.


**Experience Assessment:**

I have published one or two papers in this area.

**Review Assessment: Checking Correctness Of Derivations And Theory:**

I assessed the sensibility of the derivations and theory.

**Review Assessment: Checking Correctness Of Experiments:**

N/A

**Review Assessment: Thoroughness In Paper Reading:**

I read the paper at least twice and used my best judgement in assessing the paper.

---

> ### Author Response · Authors · 2019-11-13
> **Response to Reviewer #3 (part1)**
>
> We thank the reviewer for the detailed review and questions. We hope the following address your concerns.
>
> ## A1. The meaning of "A priori estimate" and "posterior data distribution"
>
> Thank you for pointing it out. We rewrote it to make it clearer.
>
> The word "A priori estimate" is used in the theory of partial differential equations, but already used in machine theory for example:
> Weinan E et al., A Priori Estimates for Two-layer Neural Networks (arXiv:1810.06397).
>
> A priori is Latin for "from before" and refers to the fact that the estimate for the solution of minimizing an objective function is derived before the solution is known to exist.
>
> In other words, we show that in Theorem 2, the difference between the network output function and its linear interpolation is evaluated only from the amount of the weight change. We note that in Theorem 2, we dose not use the properties of the trained neural networks.
>
> Posterior data distribution means the data distribution after training. We wanted to make clear the difference from generalization bounds in statistical learning theory, which uses certain properties of the trained neural networks.
>
> The way we use "posterior" is the same as the final paragraph in p.3 of
> Arora et al. 2019c, "Fine-Grained Analysis of Optimization and Generalization for
> Overparameterized Two-Layer Neural Networks" (arXiv:1901.08584).
>
> *All these bounds are posterior in nature they depend on certain properties of the trained neural networks. Therefore, one has to finish training a neural network to know whether it can generalize. Comparing with these results, our generalization bound only depends on training data and can be calculated without actually training the neural network.*
>
>
> ## A2. The importance of Theorem 2 and how it is related to the "implicit regularization"
>
> We discussed how Theorem 2 is related to implicit regularization in section 1, but we added more discussion in section 3.
>
> Theorem 2 estimates the difference between each neuron's output and its piecewise linear interpolation between data points.
>
> We consider the linear interpolation between two datapoints x_1 and x_2, which we denote by \(x(s) = (1−s) x_1 + s x_2 (s \in [0, 1])\).
>
> For input \(x(s)\), each layer \(l \in [L]\), each neuron \(i \in [m]\) each step of SGD \(t \in [T]\), we denote by \(A(s)\) the unit output (i.e. \(A(s) = g^{l,(t)}_{i} (x(s))\)).
>
> We denote \(B(s)\) by linear approximation of \(A(s)\) at \(s=0\) (i.e. \(B(s) = g^{l,(t)}_{i} (x(0)) + s(\frac{d}{ds}g^{l,(t)}_{i} (x(s)))\)).
>
> Then, equation (5) and (6) tell the difference between \(A(s)\) and \(B(s)\) is uniformly bounded. Furthermore, from equation (5), it can be seen that the hidden unit output is nearly equal to its piecewise linear approximation at data points when \(m\) is large enough (this has been discussed in our contributions).
>
> In other words, it can be seen that each hidden unit output connects the data points almost straight. This shows that degrees of freedom in the over-parameterized neural network is very low. The result is compatible with the NTK.
>
> We note that this estimation DOES NOT require an assumption that the weight changes follow Gaussian. We only assume that the weight changes by SGD are as small as the convergence theory (Allen-Zhu et al. 2018b) ensures.
>
>
> ## A3. Proof of equation (6)
>
> Thank you for pointing it out. It can proved by changing the last layer's output size from \(m\) to \(c\) in the proof of Theorem 2. We added a simple description in the proof of Theorem 2. Please refer to the latest version of our paper for more details.
>
>
> ## A4. To generalize our results to high-dimension regimes
>
> To generalize our results to high-dimension input cases, we have to use Rademacher complexity.
>
> What our paper claims is that a generalization error bound is relatively easily derived from implicit regularization like Theorem 2.
>
> It might be seen the trained neural networks generalize because its error from the piecewise linear interpolation is small.
>
> In Theorem 2, we prove low complexity of DNN in HIGH-DIMENSION regimes, which is derived from a mechanism of implicit regularization in DNNs.
>
> Recent works (Valle-Perez et al. ICLR 2019 and Rahaman et al. ICML 2019) have shown that the trained neural networks generalize because it is biased towards simple functions. In our paper, the bias is expressed as Theorem 2.
>
> Valle-Perez et al.,
> Deep learning generalizes because the parameter-function map is biased
> towards simple functions
> ICLR 2019.
>
> Rahaman et al.,
> On the Spectral Bias of Neural Networks
> ICML 2019.
>
> In high-dimension regimes, we cannot benefit from piecewise linear approximation. Generalizing to high-dimension regimes could be achieved by applying formula (5) to prove that the empirical Rademacher complexity of DNNs is small. It is important to generalize to high-dimension regimes, but we leave this as a suggested direction for future research.

---

> ### Author Response · Authors · 2019-11-13
> **Response to Reviewer #3 (part2)**
>
> We appreciate your detailed reading of the paper and thoughtful comments. We have responded to these below.
>
> ## A5. Contradiction(?) between random label and Theorem 4
>
> Certainly, the random label problem is important, but it does not conflict with Theorem 4.
>
> The random label method is introduced to show that the Rademacher complexity of an over-parameterized neural network is very large, and it does not lead to poor generalization ability of neural networks (Zhang et al. 2016).
>
> In our setting, the target function \(f^{*}\) is a \(C^1\)-class function and is set independently of any other settings.
>
> As is written in section 4, each data point and its label \((x_i, y_i)\) is sampled from \(f^{*}\). Theorem 4 evaluates the difference between two functions - the target function \(f^{*}\) and the neural network \(f\) trained with the dataset \(\{(x_i, y_i)\}_{i \in [n]}\).
>
> Theorem 4 can be proven from Theorem 2, calculating the difference between \(f\) and its piecewise linear interpolation between data points \(\hat{f}\).
> Then, we can understand neural networks almost straightly connect the output corresponding to the data point input.
>
> When a random label dataset is given, the neural network converges to a piecewise linear function connecting its random label output corresponding to the input.
>
> The generalization error must be high if the piecewise linear function connecting the neural network output corresponding to the random label dataset were not close to the true target function \(f^{*}\). For short, to train with random label training data means not to sample it from true target function \(f^{*}\).

---

### Decision · Program_Chairs · 2019-12-19

**Decision:**

Reject

**Comment:**

This paper analyzes a mechanism of the implicit regularization caused by nonlinearity of ReLU activation, and suggests that the learned DNNs interpolate almost linearly between data points, which leads to the low complexity solutions in the over-parameterized regime. The main objections include (1) some claims in this paper are not appropriate; (2) lack of proper comparison with prior work; and many other issues in the presentation. I agree with the reviewers’ evaluation and encourage the authors to improve this paper and resubmit to future conference.